# Enhanced polarization switching characteristics of HfO$_2$ ultrathin films via acceptor-donor co-doping

Chao Zhou[1,11], Liyang Ma[2,11], Yanpeng Feng [3,11], Chang-Yang Kuo[4,5], Yu-Chieh Ku[4], Cheng-En Liu[4], Xianlong Cheng[1], Jingxuan Li[1], Yangyang Si[1], Haoliang Huang[6], Yan Huang [1], Hongjian Zhao [7], Chun-Fu Chang [8], Sujit Das [9], Shi Liu [2] ✉ & Zuhuang Chen [1,10] ✉

In the realm of ferroelectric memories, HfO$_2$-based ferroelectrics stand out because of their exceptional CMOS compatibility and scalability. Nevertheless, their switchable polarization and switching speed are not on par with those of perovskite ferroelectrics. It is widely acknowledged that defects play a crucial role in stabilizing the metastable polar phase of HfO$_2$. Simultaneously, defects also pin the domain walls and impede the switching process, ultimately rendering the sluggish switching of HfO$_2$. Herein, we present an effective strategy involving acceptor-donor co-doping to effectively tackle this dilemma. Remarkably enhanced ferroelectricity and the fastest switching process ever reported among HfO$_2$ polar devices are observed in La$^{3+}$-Ta$^{5+}$ co-doped HfO$_2$ ultrathin films. Moreover, robust macro-electrical characteristics of co-doped films persist even at a thickness as low as 3 nm, expanding potential applications of HfO$_2$ in ultrathin devices. Our systematic investigations further demonstrate that synergistic effects of uniform microstructure and smaller switching barrier introduced by co-doping ensure the enhanced ferroelectricity and shortened switching time. The co-doping strategy offers an effective avenue to control the defect state and improve the ferroelectric properties of HfO$_2$ films.

Emerging demands for high-speed computing and high-density memory render the Von Neumann architecture less competent for future challenges. To address these issues, neuromorphic devices that mimic human synapses behaviors[1,2] along with in-memory computing technologies[3,4], have garnered significant attention. Ferroelectric memory devices, with stable memory state, low energy consumption, fast switching speed, and reliable endurance, emerge as promising candidates for these evolving technologies. Nevertheless, hindered by

[1]School of Materials Science and Engineering, Harbin Institute of Technology, Shenzhen 518055, China. [2]Key Laboratory for Quantum Materials of Zhejiang Province, Department of Physics, School of Science, Westlake University, Hangzhou, Zhejiang 310024, China. [3]Shenyang National Laboratory for Materials Science, Institute of Metal Research, Chinese Academy of Sciences, Wenhua Road 72, Shenyang 110016, China. [4]Department of Electrophysics, National Yang Ming Chiao Tung University, Hsinchu 30010, Taiwan. [5]National Synchrotron Radiation Research Center, Hsinchu, Taiwan. [6]Department of Physics, Southern University of Science and Technology, Shenzhen 518055, China. [7]Key Laboratory of Material Simulation Methods and Software of Ministry of Education, College of Physics, Jilin University, Changchun 130012, China. [8]Max-Planck Institute for Chemical Physics of Solids, Nöthnitzer Str. 40, 01187 Dresden, Germany. [9]Materials Research Centre, Indian Institute of Science, Bangalore 560012, India. [10]Flexible Printed Electronics Technology Center, Harbin Institute of Technology, Shenzhen 518055, China. [11]These authors contributed equally: Chao Zhou, Liyang Ma, Yanpeng Feng. ✉e-mail: liushi@westlake.edu.cn; zuhuang@hit.edu.cn

the size effect[5] and incompatibility with standard complementary metal oxide semiconductor (CMOS) processing, the development of ferroelectric memory devices remains challenging and progresses slowly.

The situation has changed since the discovery of ferroelectricity in commercial high-$\kappa$ HfO$_2$ thin films[6]. The fluorite-structured ferroelectric thin films have attracted extensive attention in both academia and industry due to their mature preparation process and superior CMOS compatibility. Additionally, the intriguing "reverse size effect"[7], characterized by enhanced ferroelectricity with reduced film thickness of HfO$_2$, makes it a promising candidate for the next-generation, high-density, nonvolatile memory technologies, including ferroelectric random-access memory[8], ferroelectric field-effect transistor[9], and ferroelectric tunnel junction[10]. It is widely accepted that the structural origin of ferroelectricity in HfO$_2$ films is rooted in the metastable orthorhombic phase with $Pca2_1$ space group. Defects, such as charged oxygen vacancies, are suggested to play a crucial role in stabilizing this metastable ferroelectric phase[11–13]. Acceptor-cation doping has been identified as an effective approach to introduce charged oxygen vacancies[14] and induce ferroelectricity in HfO$_2$[15,16]. In particular, trivalent rare earth dopants have shown a pronounced effect in promoting the ferroelectric phase of HfO$_2$[17]. For instance, extensive experiments have demonstrated that La-doped HfO$_2$ (La:HfO$_2$) films[18] typically exhibit appreciable polarization values compared to films with various other dopants[6,19,20]. Interestingly, there are relatively few studies on the ferroelectric properties of donor doped HfO$_2$ thin films. This is mainly because donor doping (e.g., Nb$^{5+}$ doping) would suppress oxygen vacancy formation in HfO$_2$ and thus favor stabilizing the non-ferroelectric monoclinic phase[21]. Despite the superb effect in triggering ferroelectricity, the role of oxygen vacancies remains a double-edged sword. While being effective in stabilizing the polar phase, oxygen vacancies can induce local structural/field inhomogeneities, thereby hindering domain wall motion[22,23] and thus impeding the polarization response of HfO$_2$-based devices[24]. As a result, the polarization switching characteristics of HfO$_2$ thin films often fall behind conventional perovskite ferroelectrics like Pb(Zr,Ti)O$_3$[25]. For instance, the switching time in La:HfO$_2$ films remains at microsecond level[26], nearly two orders of magnitude longer than that of Pb(Zr, Ti)O$_3$, limiting the application of HfO$_2$-based ferroelectrics in ultra-fast memory devices.

Understanding and enhancing the polarization switching properties of ferroelectric HfO$_2$ thin films are crucial for memory applications. However, achieving both a large switchable polarization and a short switching time simultaneously in HfO$_2$ films remains a challenging task. In response to the contradictory effect induced by acceptor-doping, we propose a donor (Ta$^{5+}$)-acceptor (La$^{3+}$) co-doping method to address this dilemma. We have identified factors affecting the switching characteristics of doped HfO$_2$ films from perspectives of defective state, microstructure, lattice configuration and switching path. Benefiting from the refined defect state regulated by the co-doping strategy, the La$^{3+}$-Ta$^{5+}$ co-doped HfO$_2$ (LT:HfO$_2$) films exhibit a significantly more organized and less flawed microstructure, along with a reduced switching barrier simultaneously. A remarkable 83% increase in remanent polarization ($P_r$) and a reduction in coercive fields have been achieved for the co-doped HfO$_2$ samples. Moreover, robust macro-electrical characteristics of the co-doped HfO$_2$ devices endure even in films as thin as 3 nm. Most notably, a substantial enhancement in switching kinetics has been achieved, manifesting as the impressively fastest switching process ever reported in HfO$_2$ based ferroelectric devices.

## Results
### Structural and defective state characterizations
HfO$_2$-based films with thicknesses ranging from 3 to 10 nm were deposited on (001)-oriented La$_{0.67}$Sr$_{0.33}$MnO$_3$ (LSMO)-buffered SrTiO$_3$

(STO) substrates by pulsed-laser deposition (Methods and Supplementary Fig. S1a). Figure 1a displays the representative XRD $\theta$–$2\theta$ pattern of 6 nm-thick HfO$_2$, 2% La:HfO$_2$, 2% Ta-doped HfO$_2$ (Ta:HfO$_2$), and 2% LT:HfO$_2$ films. Besides the (00 $l$) reflections of the STO substrates and LSMO electrodes, peaks locating at ~30° in all films correspond to the (111)$_O$ reflection of ferroelectric orthorhombic phase of HfO$_2$. While the additional peak appearing around 28° in HfO$_2$ and Ta:HfO$_2$ films can be assigned to ($\bar{1}$11)$_m$ plane of the paraelectric monoclinic phase. Remarkably, clear Laue oscillations near the HfO$_2$ (111)$_O$ diffraction peak is observed in the co-doped LT:HfO$_2$ film, indicating the high crystalline quality and smooth interfaces. The high crystalline quality of the LT:HfO$_2$ film is further supported by X-ray rocking curve studies (Supplementary Fig. S1b). The XRD studies clearly demonstrated that La:HfO$_2$ and LT:HfO$_2$ films have a higher fraction of ferroelectric phase than HfO$_2$ and Ta:HfO$_2$ films. Acceptor La$^{3+}$ doping effectively stabilizes the metastable polar phase, consistent with previous reports[15,27]. In contrast, donor Ta$^{5+}$ doping doesn't seem to be an effective polar structure stabilizer, as the low energy monoclinic phase is more prone to exist. All told, LT co-doping inherits the merits of La$^{3+}$ doping while overcoming the weakness of Ta$^{5+}$ doping.

Furthermore, linearly polarized X-ray absorption spectroscopy (XAS) was employed to investigate the electronic structure of the films. O-$K$ edge XAS reflects an unoccupied O $2p$ electronic density of states with an O $1s$ core hole. In transition metal oxides, the O $2p$ states are significantly hybridized with the orbital states of the metal ions through the metal-oxygen orbital hybridization. In Fig. 1b, the O $K$-edge XAS of the films is depicted. The spectral weight within the pre-edge region is primarily determined by the number of holes in the O $2p$ states, projected onto the ligand orbitals[28]. These orbitals exhibit precise symmetry bonding with the Hf $5d$ orbitals. As demonstrated in Fig. 1b, the spectral weight near the pre-edge region for both La:HfO$_2$ and LT:HfO$_2$ films can be divided into two distinct manifolds, namely $t_{2g}$ and $e_g$, due to the crystal field splitting effect. Furthermore, the X-ray Linear Dichroism (XLD, Fig. 1b) is observed as the spectral weight difference of the XAS measurements taken with the polarization vector **E** of the incoming X-rays near parallel (**E**//$c'$) and perpendicular (**E**//$ab$) to the surface normal of the film. The observation of XLD unveils an anisotropy between the out-of-plane Hf $5d$-O $2p$ bonding and the in-plane Hf $5d$-O $2p$ bonding. This directly indicates the presence of a polar rhombic pyramidal distortion in both the La:HfO$_2$ and LT:HfO$_2$ films[7], but differs in size. Notably, the dichroism signal of the co-doped LT:HfO$_2$ film exhibits a more pronounced characteristic compared to that of the La$^{3+}$ doped sample. This implies an amplified oxygen polyhedral distortion, indicating enhanced polarization in the co-doped film.

To gain a deeper understanding of the polar phase emerging in ferroelectric LT:HfO$_2$ films, XRD pole figure measurement was carried out. Interestingly, as shown in Fig. 1c and Supplementary Fig. S2b, twelve diffraction spots appeared at $\chi \approx 56°$ and 71°, respectively for {002} and {111} planes. As indicated by the colored circles in Fig. 1c and Supplementary Fig. S2b, there are four domain variants (marked as A, B, C and D) in LT:HfO$_2$ films grown on LSMO buffered STO (001) substrates and each domain variant is rotated 90° in-plane with respect to each other. This rotation is attributed to the four-fold symmetry of (001)-oriented cubic substrates. However, as illustrated by the STEM results in Fig. 1f, g and Supplementary Fig. S3, domains do not strictly align by rotating 90° in sequence. As schematically shown in Fig. 1e, the four domain variants rotate randomly in plane, leading to various configurations such as ABCD, ABBB, ADAC, ACBD, BCDA, AABB, BBBB, and so on. Additionally, $2\theta$ scans were conducted for the diffraction points observed in pole figures. As displayed in Fig. 1d and Supplementary Fig. S2c, the obtained results indicate a rhombohedral distortion of the polar phase. Considering the orthorhombic characteristic of freestanding HfO$_2$ ferroelectric films[29], the polar phase

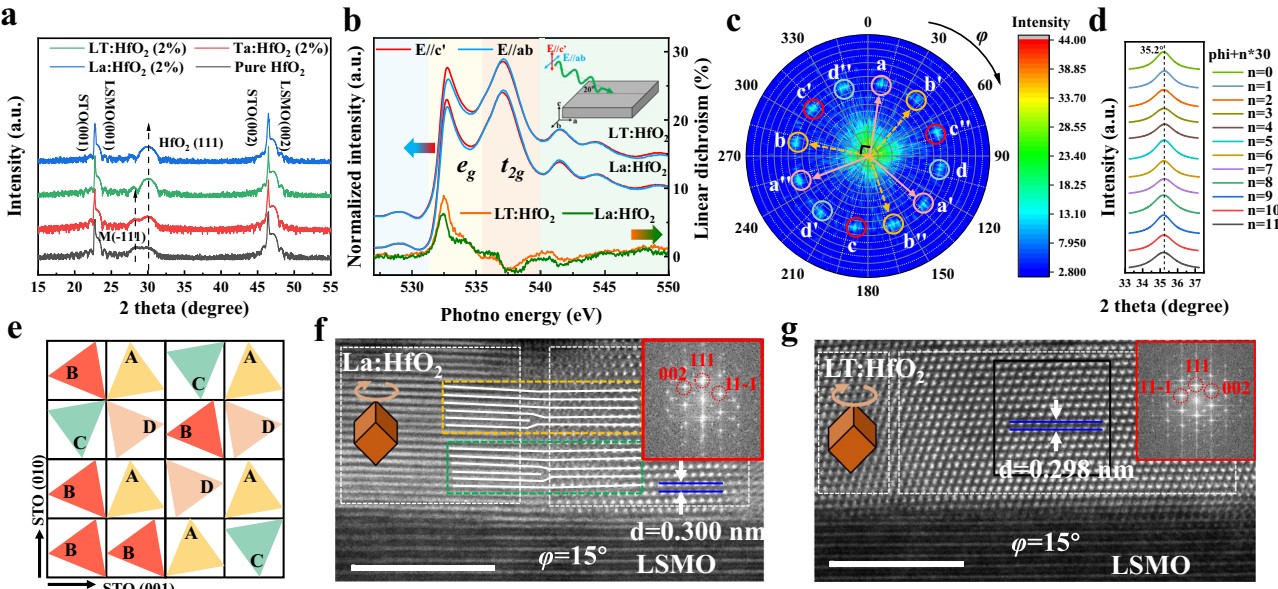

**Fig. 1 | Structure characterizations and domain distribution of HfO₂-based films. a** XRD $\theta{-}2\theta$ patterns of the films. **b** XAS and XLD at O-$K$ edge of La³⁺ doped and LT co-doped samples. The shaded background regions with different colors represent different crystal fields, and the inset shows a schematic of the experimental configuration for the spectroscopy studies. **c** The pole figure of {002} planes observed along [111] direction of the LT:HfO₂ film. The colored cycles in the pole figure represent 4 domain variants marked as A, B, C and D types, where the A type domain includes a (002), a' (020) and a" (200) planes (Likewise, B type: b (002), b' (020) and b" (200) planes; C type: c (002), c' (020) and c" (200) planes; D type: d (002), d' (020) and d" (200) planes. **d** $2\theta$ scans of 12 diffraction points in the{002}-plane pole figure. The displayed $\theta{-}2\theta$ data were processed with Lorentz fitting. **e** An example of HfO₂ (111) domain distribution on the (001) plane of 4 × 4 STO supercells. STEM images observed along $\varphi = 15°$ (zone axis STO [001] defined as $\varphi = 0°$) of (**f**) the La:HfO₂ film and (**g**) the LT:HfO₂ film. The rhombohedrons inset indicate the domains within HfO₂ films with different in-plane rotation angles. The white solid lines enclosed with green and yellow dotted lines in **f** describe the dislocations.

appearing in the HfO₂/LSMO/STO (001) system can be defined as a orthorhombic phase with a small rhombohedral distortion.

Cross-sectional high-angle annular dark-field (HAADF) STEM images offer a detailed insight into the microstructure of doped HfO₂ films. As seen in Fig. 1f, g, various domain variants are visible, displaying fringes and lattice forms in localized regions. The observed results are consistent with pole figures, indicating the distribution of domains with different in-plane rotation angles. As shown in Fig. 1f and Supplementary Fig. S4, numerous dislocations are present within domains and at domain boundaries in the La:HfO₂ sample. Moreover, when examining La:HfO₂ films from a larger area scale perspective (Supplementary Fig. S3e), additional variations like domain tilt and domain misalignment become apparent. In contrast, the situation in the LT:HfO₂ is considerably better than that of the La:HfO₂ films. As illustrated in Fig. 1g and Supplementary Fig. S3a, coherent domain boundaries and a uniform 0.298 nm $d$-spacing of (111) planes are evident in the LT:HfO₂ sample. Moreover, the LT:HfO₂ films exhibit a distinctly lower dislocation density compared to that of the La:HfO₂ films, as shown in Supplementary Figs. S4 and S5. Considering the identical growth condition of these doped HfO₂ films, the increased occurrence of dislocations in La:HfO₂ films is likely associated with a higher concentration of oxygen vacancies, which usually promotes the nucleation of dislocations[30,31]. In a word, the microstructure of LT:HfO₂ films is highly ordered, and dislocations are rare in comparison to La:HfO₂.

To investigate the doping effect on the valence and defect states, X-ray photoelectron spectroscopy (XPS) studies were conducted (Fig. 2). The core level of Hf $4f_{7/2}$ from the undoped HfO₂ film is located at 16.60 eV. Interestingly, the acceptor La³⁺ doped and donor Ta⁵⁺ doped films exhibit the contrary chemical shifts of the Hf $4f$ core levels. In comparison to undoped HfO₂, the Hf $4f_{7/2}$ binding energy of the Ta:HfO₂ shifts to a higher region (16.75 eV), while that of the La:HfO₂ moves to a slightly lower position (16.50 eV). The O $1s$ spectra

of La:HfO₂ and Ta:HfO₂ films present the same shift trend (Supplementary Fig. S6). Similar phenomena have been observed in oxides that undergo reduction or oxidation process[32–34]. Meanwhile, different chemical shifts observed in Ta $4f$ and La $3d$ (Fig. 2b, c) peaks rule out the influence of work function changes caused by doping on the binding energy. According to the defect equation of La³⁺ doping ($La_2O_3 \xrightarrow{2HfO_2} 2La'_{Hf} + V_{\ddot{O}} + 3O_O$), oxygen vacancies are introduced to maintain charge neutrality, resulting in a reduced environment. Similarly, the reverse effect occurs when Ta⁵⁺ acts as a dopant in HfO₂. In the case of the Ta:HfO₂ film, the extra oxygen could mitigate the effect of oxygen vacancies. Consequently, the co-doping of La³⁺ and Ta⁵⁺ neutralizes the influence of single component doping, and the Hf $4f$ signals virtually return to almost the same position (16.62 eV) as that of undoped HfO₂. Moreover, the presence of Hf $4f$ signals from HfO₂-ₓ suboxide serves as direct evidence of the defect condition after doping. As shown, the suboxide signal is more intense in La³⁺ doped HfO₂ due to the introduction of more oxygen vacancies; yet, Ta⁵⁺ doping has the opposite effect, leading to the nearly-dismissed Hf $4f$ signature of suboxide. Thus, the co-doping method provides a simple but effective way to regulate the oxygen content and oxygen vacancies in doped-HfO₂ films.

## Ferroelectric properties

The electrical performances of HfO₂ films with different doping types were further characterized. Polarization-electric field loops and corresponding I-V results of 6-nm-thick films with different doping types are depicted in Fig. 3a, b. The pure HfO₂ film exhibits weak ferroelectricity, evident from the faint current peaks during polarization switching. The situation does not improve for the Ta⁵⁺ doped film; in fact, the leakage current even increases, as shown in Supplementary Figs. S7h and S8c. Consistent with XRD results, substantial improvement of ferroelectricity is achieved in both La:HfO₂ and LT:HfO₂ films. However, the polarization value of La:HfO₂ is lower, and its switching

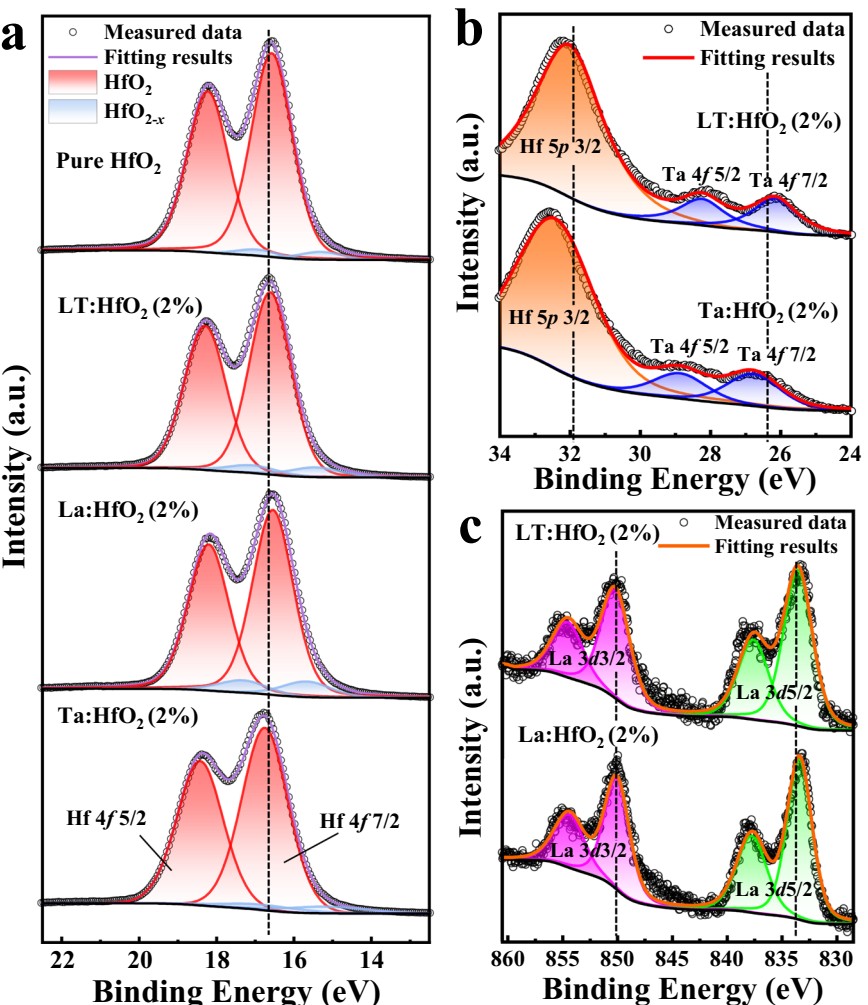

**Fig. 2 | XPS of the 6-nm-thick HfO$_2$ films with different doping conditions. a** Hf 4$f$ spectra comparing pure HfO$_2$ and HfO$_2$-based films with different doping states. **b** Ta 4$f$ and Hf 5$p$ spectra of LT:HfO$_2$ and Ta:HfO$_2$ films. **c** La 3$d$ spectra of La:HfO$_2$ and LT:HfO$_2$ films.

current peaks appear blunter compared to those of LT:HfO$_2$. This can be ascribed to a higher defect density, including oxygen vacancies, dislocations, and microstructure distortions[35–37], which hinder the domain wall motion and consequently result in the "blunt" switching current peaks in La:HfO$_2$ films. In other words, the speed of polarization reversal cannot keep up with the change of the external electric field, leading to relatively lower polarizations. In contrast, the co-doped film exhibits a significantly enhanced polarization with sharper switching current peaks, which benefits from the uniform microstructure with fewer defects. The 2% LT:HfO$_2$ film has a 2$P_r$ of 38 μC cm$^{-2}$, which is 83% higher than that of the La:HfO$_2$ film, consistent with aforementioned XLD results. Moreover, the coercive field $E_C$ of the co-doped film decreases. Furthermore, a significant enhancement in remnant polarization values is observed in all co-doped samples (Fig. 3c and Supplementary Fig. S7c, S7d), regardless of the doping levels. LT co-doped samples with varying film thicknesses are further characterized (Fig. 3d, Supplementary Fig. S7g and S7i). Robust and reliable ferroelectricity is identified in the LT:HfO$_2$ films even down to a thickness of just 3 nm. This result signifies the high film quality and less defective microstructure achieved through the co-doping method, which guarantees less resistive switching current in the ultrathin wide bandgap material (4.41 eV for $Pca2_1$ HfO$_2$[38]). This ensures the macroscopic ferroelectricity of the 3 nm thick sample. Notably, to our best knowledge, this is one of the thinnest films whose reliable macroscopic ferroelectricity can be directly obtained[39,40]. Meanwhile, the leakage

current and endurance behaviors of these LT:HfO$_2$ thin films remain comparable to those of La:HfO$_2$ films (Supplementary Figs. S7h, S7i, S8). Altogether, these findings highlight that the co-doping strategy not only effectively enhances the ferroelectric performances of HfO$_2$ films, but also elevates the quality of ultrathin films, which expand the range of possibilities for ultrathin devices.

## Switching dynamics characterizations

As depicted above, the presence of "sharper" switching current peaks and reduced $E_C$ collectively indicate the faster switching dynamics observed in the co-doped films. Therefore, the switching kinetics of doped-HfO$_2$ films are investigated via pulse switching measurements using a designed pulse train sequence illustrated in Supplementary Fig. S9. Figure 4a, b displays the switched polarization fraction as a function of time at varying external fields for LT:HfO$_2$ and La:HfO$_2$ films, respectively, tested on top electrodes with a diameter of 50 μm ($d$ = 50 μm). Overall, the co-doped film exhibits a notably faster switching behavior than the La:HfO$_2$ films. In specific, a complete switching takes about 2000 ns for the La:HfO$_2$ device, whereas it just spends less than 700 ns for the co-doped sample.

Polarization switching in ferroelectrics involves the nucleation of the reversed domain followed by the domain growth. Two prevalent models, the nucleation limited switching (NLS) model[41,42] and Kolmogorov-Avrami-Ishibashi (KAI) model[43,44] are commonly used to interpret the switching kinetics of ferroelectrics. The NLS model,

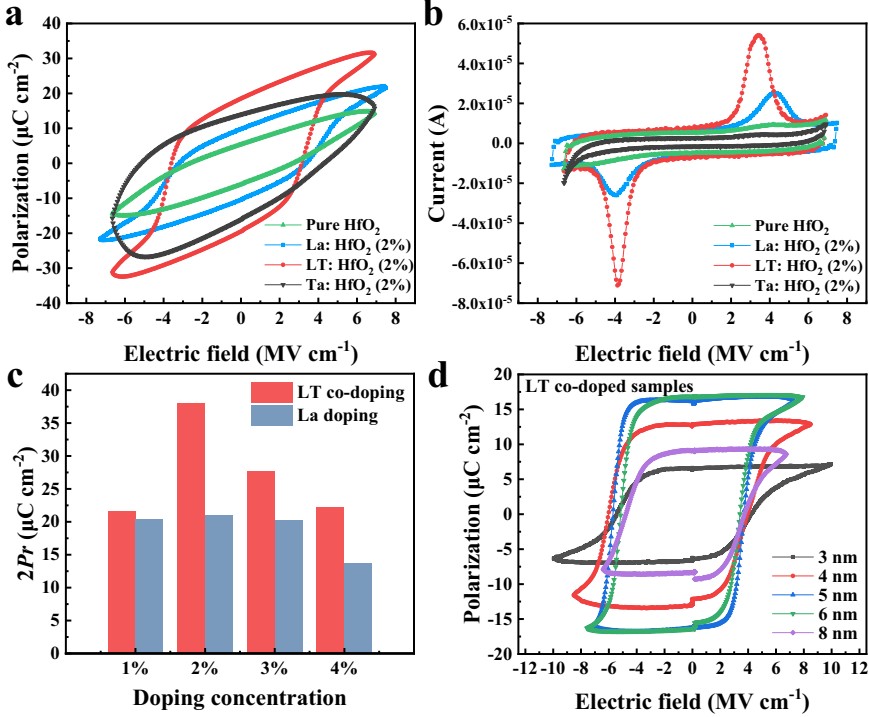

**Fig. 3 | Electric measurements of doped HfO₂ films. a** P-E loops and (**b**) corresponding I-V curves of 6-nm-thick HfO₂ films with different doping conditions. **c** Comparison of remnant polarization values of 6-nm-thick La³⁺ doped and LT co-doped HfO₂ films with different doping concentrations. **d** PUND curves of LT co-doped HfO₂ films with different thicknesses.

which assumes that the polarization switching is limited by the domain nucleation, is often employed to explain inhomogeneous systems such as ferroelectric ceramics and polycrystalline films[42]. On the other hand, the KAI model, based on the assumption that the switching is dominated by the domain growth, has been successfully used to describe domain switching kinetics in homogeneous system, such as single crystals and epitaxial films. Both models can be interpreted by the following equation:

$$\Delta P(t) = 2P_S \int_{-\infty}^{+\infty} \left[ 1 - \exp\left\{ -\left(\frac{t}{t_0}\right)^n \right\} \right] F(\log t_0)\mathrm{d}(\log t_0) \quad (1)$$

Where $P_S$ is the spontaneous polarization, $F(\log t_0)$ is a probability distribution function of the characteristic switching time $t_0$, and $n$ is the effective dimension of domain growth. The $F(\log t_0)$ is specific for different models. For the NLS model, influenced by defects such as oxygen vacancies, grain boundaries, and impurities, the nucleation of reversed domains does not take concerted manner and thus the distribution function is a Lorentzian function:

$$F(\log t_0) = \frac{A}{\pi} \left[ \frac{\omega}{(\log t_0 - \log t_1)^2 + \omega^2} \right] \quad (2)$$

Where $A$ is a normalized constant; $\omega$ and $\log t_1$ are the half-width at half-maximum and center of the distribution, as presented in the inset of Fig. 4e. While for the KAI model, $\omega$ approaches zero and the distribution function $F(\log t_0)$ becomes a Dirac-delta function and Eq. (1) of this model can be written as:

$$\Delta P(t) = 2P_S \left[ 1 - \exp\left\{ -\left(\frac{t}{t_0}\right)^n \right\} \right] \quad (3)$$

Therefore, the KAI model can be considered as a special case of the NLS model. Consequently, both the NLS model and KAI model are

employed to fit the measured data. As demonstrated in Fig. 4a, b, only the NLS model yields a satisfactory fit for the switching kinetics of the La:HfO₂ film. Conversely, the switching kinetics of the LT:HfO₂ film can be equally well described by both the NLS and KAI models (above 5 MV cm⁻¹). The Lorentzian distribution functions used to fit the switching kinetics of the LT:HfO₂ and La:HfO₂ films are presented in Fig. 4c, d, respectively. With increasing applied fields, $\log t_1$ decreases, and the Lorentzian distribution functions become sharper, indicating that defects, impurities, and other local potential variations of inhomogeneous systems are smoothed out under sufficiently large electric fields. Figure 4e illustrates the trend of $\omega$ as a function of electric fields, where the co-doped films consistently show smaller $\omega$ across all fields. In the case of co-doped LT:HfO₂ films, the $\omega$ approaches zero when the electric field exceeds 5 MV cm⁻¹, suggesting the validity of the KAI model. In contrast, the continuous change of $\omega$ with increasing applied electric fields suggests an NLS-type switching for La:HfO₂ films. Concurrently, impacts of the imprint phenomenon on the switching dynamics are also investigated (Supplementary Fig. S10). A similar changing tendency of the switching model is observed in the negatively poled region, proving the validity of our experiment. The change in the switching mechanism further confirms a homogeneous and less defective microstructure achieved through the co-doping method. In addition, faster kinetics can be achieved with thicker films or when the capacitor areas are further minimized[45–47]. Thus, as highlighted in Supplementary Fig. S11, the switching dynamics become faster as electrode areas are reduced. For the 6-nm-thick co-doped LT: HfO₂ film, a full switching completes within 78 ns as measured on a capacitor with an electrode diameter of $d = 12.5$ μm. Accordingly, the switching speed is further enhanced in thicker films, which is greatly attributed to the reduced depolarization fields for thicker samples. For example, a 10-nm-thick LT: HfO₂ film completes its switching within 50 ns (tested on a $d = 12.5$ μm electrode). Therefore, it is predicted that the switching time of the films could be further minimized to the sub-ns range[45,47], as summarized in Fig. 4f [26,45,47–57]. The fit lines (dashed lines)

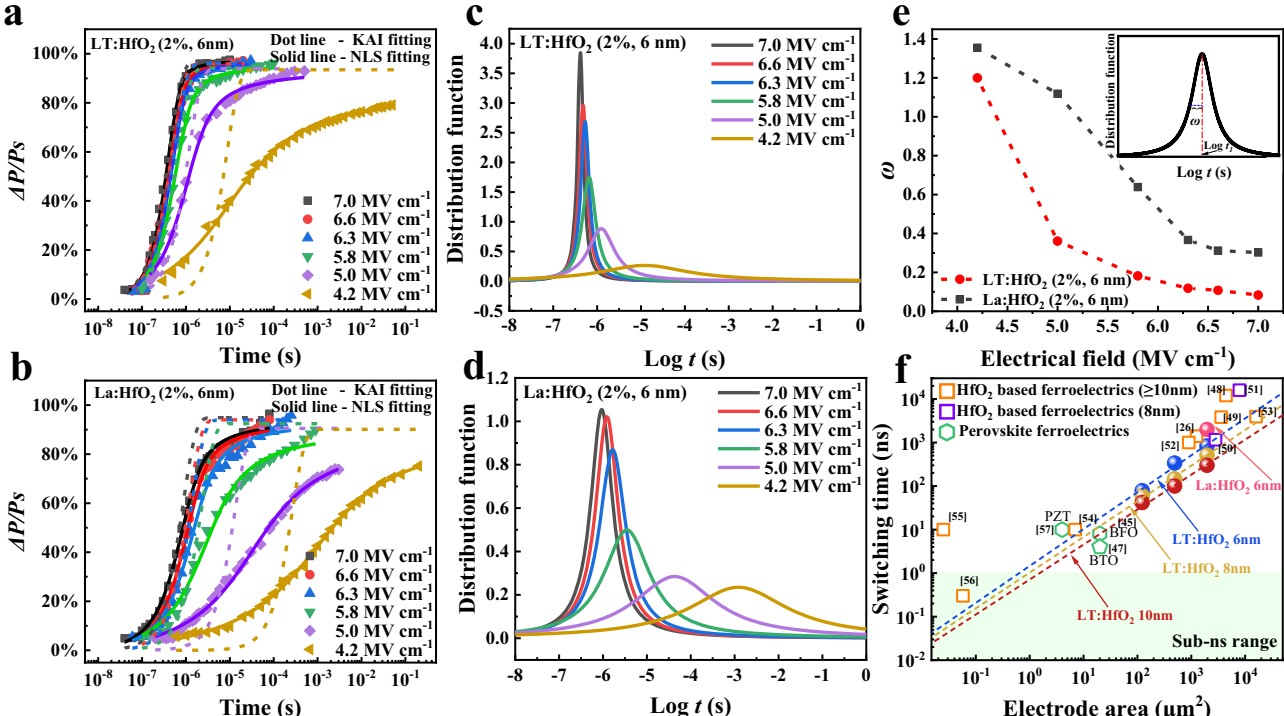

**Fig. 4 | Switching dynamics characterizations of La:HfO₂ and LT:HfO₂ films.** Switchable polarization measured with different write amplitudes and duration time of the 6-nm-thick (**a**) LT: HfO₂ and (**b**) La: HfO₂ films. The dotted lines and solid lines in those pictures represent the KAI and the NLS model fitting results. Lorentzian distribution functions of 6 nm (**c**) LT:HfO₂ and (**d**) La:HfO₂ films. **e** The changing tendency of ω as a function of applied electric fields. **f** Demonstration of the switching time of La:HfO₂ and LT:HfO₂ capacitors with different thicknesses and electrode areas and comparison with the switching time of other ferroelectric devices. Dashed lines are the linear fitting results of our devices and the shaded region represents the sub-nano-second switching time region.

exhibit the evolution of switching time versus capacitor area and indicate that the 10-nm-thick LT:HfO₂ devices embody sub-ns switching with a capacitor area of ~2 μm². The switching speed of LT:HfO₂ films, as presented in this study, stands out as the top among the reported doped HfO₂ systems by linear extrapolation and is comparable with that of perovskite-structured ferroelectrics. All told, our results unequivocally demonstrate that the co-doping strategy stands as an effective method to enhance the switching behaviors of ferroelectric HfO₂ thin films.

## Atomistic insight into the switching performances in HfO₂

Density functional theory (DFT) calculations were carried out to provide atomistic insights into the structural properties of ferroelectric HfO₂ with various doping types and to elucidate the mechanism underlying the enhanced switching properties in co-doped films. As detailed in the Supplementary Figs. S13–S15, a $2 \times 2 \times 1$ supercell is used to model HfO₂ systems with different doping types. For La-doped HfO₂, we discover that the lowest-energy configuration has a three-fold coordinated (positively charged) oxygen vacancy near two La³⁺ ions substituting two Hf⁴⁺ ions. The presence of charged oxygen vacancy ensures the charge neutrality. In contrast, the lowest-energy configuration for LT:HfO₂ features neighboring La³⁺ and Ta⁵⁺ ions, each replacing a Hf⁴⁺ ion. Noteworthy is that there exist multiple switching pathways in ferroelectric HfO₂[58,59]. As shown in Supplementary Figs. S16–S18, the SI−2 switching routine is chosen as the pathway for the polarization reversal. The pathway with the lowest barrier involves concerted movements of three-fold coordinated (polar) and four-fold coordinated (nonpolar) oxygen ions against the direction of the applied electric field. As a result, the coordination number of oxygen ions changes: the initially three-fold coordinated polar oxygen ions become four-fold coordinated, whereas the four-fold coordinated nonpolar oxygen ions transit to three-fold coordination. Interestingly,

we find that the polarization switching in La:HfO₂ (Fig. 5a) is coupled to the migration of a three-fold coordinated oxygen vacancy, a process that is likely to be rather slow kinetically. The migration of the oxygen vacancy retains its three-fold coordination, as previous DFT studies have demonstrated that the charged oxygen vacancy strongly favors three-fold coordination[11]. In comparison, the switching process in LT:HfO₂ does not involve the slow migration of vacancies. Additionally, La³⁺-Ta⁵⁺ co-doping lowers the energy of the intermediate state (*Pbcn*-like, Fig. 5b), thereby reducing the switching barrier. These results suggest that the co-doping strategy reduces the coercive field by mitigating the adverse impact of charged oxygen vacancies on polarization switching. At the same time, switching barriers of models with different La/Ta mole ratios and supercells (indicating different doping concentrations) are also calculated. As can be seen in Supplementary Figs. S19, S20, the La/Ta model with a 1:1 ratio facilitates the switching process than that of the La single doping, which is consistent with the experiment results.

## Discussion

In summary, we have systematically explored the effects of acceptor-donor co-doping on the microstructures, ferroelectric properties, and switching kinetics of ultrathin HfO₂ films. Compared with La³⁺ doped films, La³⁺-Ta⁵⁺ co-doped HfO₂ films exhibit a notably more uniform and less defective microstructures, changing the switching mechanism from nucleation limited switching to Kolmogorov-Avrami-Ishibashi model. First-principles calculations reveal at the atomistic level that co-doping could inhibit the generation of oxygen vacancies and reduce the polarization switching barrier. These synergistic effects of co-doping manifest in the reduction of switching barriers, leading to reduced coercive fields, enhanced polarization values, shortened characteristic switching time and faster switching speed. Furthermore, the reduced defects in co-doped HfO₂ enable the production of high-

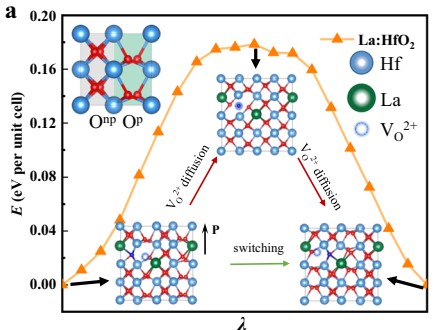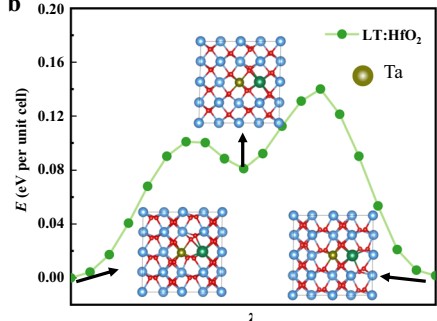

**Fig. 5 | Energy profiles for polarization switching and switching paths of doped HfO₂ in a 2×2×1 supercell. a** The energy landscape for polar switching of the La-HfO₂ system which contains 2 La cations and 1 oxygen vacancy. Insets show the nonpolar $O^{2-}(O^{np})$ and polar $O^{2-}(O^p)$ in a HfO₂ unit cell and the original, intermediate and final configurations of La-HfO₂ during the polarization switching. **b** The switching energy profile of the co-doped HfO₂ sample containing a La cation and a Ta cation. Insets present the switching pathway for the co-doped HfO₂.

quality films, which exhibit excellent macro-electrical performances even at a thickness of 3 nm, paving the way for advanced ultra-thin devices. The defect-engineering strategy provides an effective avenue to regulate the defect state and subsequently improve ferroelectric properties of HfO₂-based devices.

## Methods

### Thin-film growth

HfO₂-based films were grown on La₀.₆₇Sr₀.₃₃MnO₃ buffered-SrTiO₃ (001) substrates by pulsed laser deposition (Arrayed Materials RP-B) using a KrF excimer laser ($\lambda = 248$ nm). The thicknesses of these films were determined by the X-ray reflection (XRR) technique (Supplementary Fig. S1). The LSMO was deposited with a laser (3 HZ, 0.85 J cm⁻²) under an oxygen partial pressure of 20 Pa at the substrate temperature of 700 °C. Subsequently, the HfO₂-based films (including the pure HfO₂ and doped HfO₂ samples, i.e., La:HfO₂, Ta:HfO₂ and LT:HfO₂ samples) were grown at a temperature of 600 °C in a dynamic oxygen pressure of 15 Pa at a laser repetition rate of 2 Hz and a laser fluence of 1.35 J cm⁻². Then, the heterostructures were cooled down to room temperature at the rate of 10 °C min⁻¹ under an oxygen pressure of 10000 Pa. In addition, the composition of HfO₂-based films with different doping concentration is controlled by stoichiometric PLD targets. For instance, the 2% La:HfO₂ films are prepared using the Hf₀.₉₈La₀.₀₂O₁.₉₉ ceramic target, and the 2% LT:HfO₂ films are synthesized with the Hf₀.₉₆La₀.₀₂Ta₀.₀₂O₂ (i.e., the La and Ta are co-doped in a 1:1 mole ratio) target. All the ceramic targets were synthesized at 1500 °C via the solid-state reaction using HfO₂ (99.99% purity), La₂O₃ (99.99% purity) and Ta₂O₅ (99.99% purity) powders. The top electrode Pt [ZhongNuo Advanced Material (Beijing) Technology] with a thickness of 100 nm was patterned by photolithography and deposited by magnetron sputtering (Arrayed Materials RS-M).

### Structural analysis

Structural characterization was performed by high-resolution X-ray diffraction with Cu $K_{\alpha 1}$ radiation using a Rigaku Smartlab (9 kW) diffractometer. Two scanning modes were employed to investigate the crystal structure of HfO₂ films. One is the radial scanning mode, in which the incident and detector move with $\theta-2\theta$, that used to analyze the orientation and lattice constants of films. The other is the pole figure mode that tilts the sample plane along $\chi$ direction and rotates the sample azimuthal angle $\varphi$ with the $\theta-2\theta$ position fixed. The pole figure method offers the symmetrical characteristics along certain zone axis.

### X-ray photoelectron spectroscopy (XPS) measurements

The XPS experiments were performed using an ESCALAB 250Xi instrument (Thermo Fisher) with the Al $K_\alpha$ radiation (12 kV) as the

excitation source. In the XPS analysis, energy calibration was carried out by using C 1$s$ peak at 284.8 eV.

### Soft X-ray spectroscopy measurements

Soft x-ray spectroscopy measurements were achieved in the total electron yield mode (TEY) at the TLS11A and TPS45A beamline of the National Synchrotron Radiation Research Center (NSRRC) in Taiwan. Both the X-ray absorption spectroscopy (XAS) and the X-ray linear dichroism (XLD) measurements were conducted at room temperature without applied magnetic field using TEY mode. The incident angle of soft x-ray is 20° from the sample surface in XAS experiments. XLD signals were obtained from the difference of the horizontal (**E**//ab) and vertical (**E**//c′, c′ is the axis that 20° away from surface normal c) linearly polarized light absorption spectra. i.e., the polarization dependent XAS signals were guaranteed from a same sample area.

### Electron microscopy characterizations

The cross-sectional samples for STEM observations were prepared by conventional method of gluing, grinding, dimpling and ion milling. A PIPS 695 (Gatan) was used for final ion milling. At the beginning of ion milling, a voltage of 4.5 kV and an incident angle of 8° were performed to ion milled. Then, the incident angle was gradually reduced to 5°. The final voltage of ion milling was less than 0.5 kV to clean the surface damage by ion beam. Cross-sectional HAADF-STEM images and EDS mappings were acquired by Spectra 300 X-FEG aberration-corrected scanning transmission electron microscope (ThermoFisher Scientific) with double aberration (Cs) correctors and a monochromator operating at 300 kV.

### Electrical measurements

Electrical measurements were implemented via a metal-ferroelectric-metal (MFM) capacitance structure using aixacct TF3000 analyzer. The MFM capacitances were applied to electric fields with the Pt electrode connecting to the positive bias and La₀.₆₇Sr₀.₃₃MnO₃ grounded. Diverse measurements were used to reveal macroscopic electrical performances, including the positive-up-negative-down (PUND) measurement, leakage current, capacitance-voltage (CV), switching dynamics, endurance, and polarization-electric field (P-E) loops et al. Both the PUND measurement and P-E loops were measured with a 5 kHz triangle electric pulse while the PUND measurement subtract non-switching contributions, such as dielectric and leakage current contributions.

### First-principles calculations

All first-principles density functional theory calculations were performed using Vienna Ab initio Simulation Package (VASP) with

generalized gradient approximation of the Perdew-Burke-ErnZerhof (PBE) type. The polarization switching process in $HfO_2$-based materials is modeled using a periodically repeated 2×2×1 supercell which contains 16 Hf and 32 O atoms. The considered polarization switching cases reported in this paper include: A. La-doped $HfO_2$, in which two Hf atoms are replaced by La and an oxygen vacancy is introduced; B. La and Ta co-doping, in which two Hf atoms are replaced by a La and a Ta atom respectively, without introducing oxygen vacancy. The minimum energy paths of polarization switching in polar orthorhombic $HfO_2$ under different doping conditions are determined using the variable-cell nudged elastic band (VC-NEB) technique implemented in the USPEX code with lattice constants relaxed during polarization switching. The plane-wave cutoff is set to 600 eV. We use a 2×2×4 Monkhorst-Pack k-point grid for structural optimizations and VC-NEB calculations.

In order to avoid the influence caused by excessive doping concentration, we have carried out the test calculation for the system of larger supercells (see Supplementary Information). We show the DFT minimum energy paths for polarization switching in ferroelectric hafnia with different proportions of La/Ta (1:2, 2:1 and 1:1) in 2×2×2 supercell. The plane-wave cutoff is set to 550 eV. Here we use a 2×2×2 Monkhorst-Pack k-point grid for structural optimizations and VC-NEB calculations. Additional calculations for energy barriers of SI −2 switching pathway in La:$HfO_2$ and LT:$HfO_2$ using 4×2×2 supercells show that the co-doped system has a much lower switching barrier than the La doped system. The plane-wave cutoff is set to 550 eV. Here we use a 1×2×2 Monkhorst-Pack k-point grid for structural optimizations and VC-NEB calculations. Other convergence conditions for VC-NEB calculations are the same: the root-mean-square forces on images smaller than 0.03 eV Å$^{-1}$ is the halting criteria condition for NEB calculations. The variable elastic constant scheme is used, and the spring constant between the neighboring images is set in the range of 3.0 to 6.0 eV Å$^{-2}$.

### Reporting summary

Further information on research design is available in the Nature Portfolio Reporting Summary linked to this article.

## Data availability

The source data for Figs. 1–5 in this study are provided in the Source Data file. Source data are provided with this paper.

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

## Acknowledgements

This work was supported by National Key R&D Program of China (Grant No. 2021YFA1202100), National Natural Science Foundation of China (Grant Nos. 52372105, 12361141821), Guangdong Basic and Applied Basic Research Foundation (Grant No. 2020B1515020029), Shenzhen Science and Technology Innovation project (Grant No. JCYJ20200109112829287), and Shenzhen Science and Technology Program (Grant No. KQTD20200820113045083). Z.H.C. has been supported by "the Fundamental Research Funds for the Central Universities" (Grant No. HIT.OCEF.2022038) and "Talent Recruitment Project of Guangdong" (Grant No. 2019QN01C202). M.L. and S.L. acknowledge the supports from National Natural Science Foundation of China (Grant No.12074319). Y.P.F. acknowledges the Guangdong Basic and Applied Basic Research Foundation (Nos. 2021A1515110064, 2023A1515011058). C.-Y.K. acknowledges the financial support from the Ministry of Science and Technology in Taiwan under grant nos. MOST 110–2112-M-A49-002-MY3. C.-Y.K. and C.-F.C. acknowledge support from the Max Planck-POSTECH-Hsinchu Center for Complex Phase Materials. S.D. acknowledges Science and Engineering Research Board (SRG/2022/000058) and Indian Institute of Science start up grant for financial support.

## Author contributions

Z.H.C. and C.Z. conceived and designed the research. Z.H.C. supervised this study. C.Z. fabricated the films and carried out electric measurements with the assistance of X.L.C, Y.Y.S. and J.X.L. C.Z. performed the XRD measurements with the assistance of H.L.H. Y.P.F. performed the STEM characterizations. L.Y.M. and S.L. carried out theoretical calculations. C.-Y.K., Y.-C. K., C.-E. L., and C.-F.C. performed the soft spectroscopy measurements. C.Z., L.Y.M., S.L. and Z.H.C. wrote the manuscript. Y.H., H.J.Z. and S.D. provided insights and interpretation of the results. All authors discussed the results and commented the manuscript.

## Competing interests

The authors declare no competing interests.
