## [Peer Review File · Nature Communications]

Enhanced polarization switching characteristics of HfO₂ ultrathin films via acceptor-donor co-dopingEditorial Note: Parts of this Peer Review File have been redacted as indicated to remove third-party material where no permission to publish could be obtained.

Reviewers' comments:

Reviewer #1 (Remarks to the Author):

The authors report that La/Ta-HfO₂ films can behave enhanced polarization switching behaviors due to acceptor-donor co-doping. X-ray diffraction, XPS, TEM and ferroelectric properties are investigated to verify this effective co-doping strategy. Lower concentration of oxygen vacancies can promote the synergistic improvement of ferroelectricity and higher switching speed. Although this work is interesting, the data in the present state do not fully support the claims and the novelty is insufficient, while the investigation needs to be deeper to meet the requirements of the journal. Below is a list of my concerns:

1. Acceptor-cation doping can bring out more oxygen vacancies in HfO₂-based films. But no description about donor-cation doping was given, and more examples about this method should be provided in the introduction part.
2. The authors only introduced the growth method of HfO₂, but no description about La:HfO₂, Ta:HfO₂ and La/Ta:HfO₂ was given. The detailed growth parameters for doping HfO₂ should also be provided. At the same time, how do the authors control the doping concentration, controlling target element or adjusting sputtering parameters?
3. The authors did not give the detailed percentage of La and Ta during co-doping in the manuscript. For example, what is the atomic proportion of La and Ta in the 2% La/Ta co-doping films? Why do the authors choose this proportion? Please give more convincing evidence such as energy profile.
4. Fourier transform pictures cannot be clearly seen in Figs. 1f & 1g, especially the mark of crystal orientations. Please provide pictures with better resolution.
5. Dislocations are clearly seen in La:HfO₂, which may be caused by oxygen vacancies from the authors' point. However, dislocations can be generated by various factors, such as external strain, temperature and pressure etc. As can be acknowledged that tensile stress can also exist in HfO₂-based films. Here, the authors should give more explanations to verify that dislocations are generated just because of oxygen vacancies.
6. I suggest that the authors can try to quantitatively characterize the content of oxygen vacancies by STEM, EPR or XAFS.
7. About XPS results, the authors should provide O 1s spectra of all films and give corresponding analysis.
8. The authors proposed that larger leakage current for Ta:HfO₂ is due to the enhanced ion mobility of interstitial oxygen. However, oxygen vacancies can also promote the increase of leakage current (See from DOI: 10.7567/JJAP.57.066501 and DOI: 10.1002/adfm.202104913). So, more explanations should be given for the main contribution of leakage current in this manuscript. Or how to distinguish these two contributions?
9. Figs. S4a & 4c and Figs. S4b & 4d, the variation tendency of polarization is not consistent with that of permittivity. Please explain or analyze the intrinsic reason.
10. From Fig. 4a, it can be clearly seen that NLS model can be more appropriate to describe the switching kinetics for LT:HfO₂ film. However, KAI model is not appropriate at lower electric field. The authors should give more explanations.
11. For DFT theoretical simulation of La/Ta-HfO₂, have the authors considered other conditions such as different proportion of La/Ta? Does there appear the lowest energy in that composition?
12. In the "Discussion" part, the description of "co-doping can inhibit the migration of oxygen

vacancies" is inaccurate, while the description of "co-doping can inhibit the generation of oxygen vacancies" seems to be more accurate in the light of the author's previous viewpoints.

13. In the "ferroelectric properties" section, it should be added why the ferroelectric properties of LT:HfO₂ films are substantially improved compared with that of La:HfO₂ film in the presence of a reduced concentration of oxygen vacancies.

14. In the manuscript, the authors says that "this is the thinnest film whose reliable macroscopic P-E loops can be directly obtained". However, macroscopic P-E loops were also reported in the literature (See from DOI: 10.1126/science.abm8642). Please clarify this statement.

Reviewer #2 (Remarks to the Author):

This is very interesting work about acceptor-donor co-doping in hafnia ferroelectric materials. Compared with La or Ta single element doping, the LT co-doped sample exhibited much better ferroelectricity and higher switching speed. These features are very important for practical memory applications. I think the overall experimental results is quite clear, however, the mechanism part still has room to improve. The followings are my comments and suggestion:

1. The authors claimed the LT doped sample have much fewer defects and dislocations in structure, based on the TEM data. I suggest to check more area and larger scale in TEM. Including statistical data will make this point more solid.
2. In various literatures, La doping is an effective method to improve the HZO ferroelectricity with higher Pr, lower Ec and better endurance. However, the La doped sample did not show the same case. The process of La doping in this work may not be in the right region.
3. The claim of '3nm film is the thinnest film whose reliable macroscopic P-E loops can be directly obtained.' may not be appropriate. To my knowledge, ferroelectric of 1.5 nm thick HZO was directly measured by PUND in ref(IEEE Electron Device Letters 42 (9), 1303-1306).
4. Pls show more evidence and analysis on the switching pathway of LT doped sample. It is far more than enough in current version.
5. What is the series resistance of FeCAP, including LSMO and testing setup, the switching speed should be closely related with the series resistance. Pls also estimate the domain wall growth speed of the LT sample.

Reviewer #3 (Remarks to the Author):

Review on the manuscript "Enhanced polarization switching characteristics of HfO₂ ultrathin films Via acceptor-donor co-doping" by Chao Zhou et al. submitted to Nature Communications (NCOMMS-23-43651-T)

Zhou et al. reported the enhanced ferroelectricity in HfO₂-based ultrathin films achieved through the acceptor-donor co-doping strategy in this manuscript. The thin films are prepared using pulsed-laser deposition on La_{0.67}Sr_{0.33}MnO₃ (LSMO)-buffered SrTiO₃ (STO) substrates. By proposing a donor (Ta⁵⁺)-acceptor (La³⁺) co-doping method, the authors reported an 83% increase in remanent polarization and robust ferroelectricity compared to the La-doped HfO₂ films, which were due to the more homogeneous and less defective microstructures. In addition, the switching kinetics measurement and first-principle calculation of the switching barrier are also performed. Overall, the paper presents the in-

depth structure analysis results. Nonetheless, the technical improvement in the material properties over the previous works is marginal, significantly undermining the novelty of the work. Therefore, this work is suitable for more specialized journals. The detailed comments are appended below.

1. In Fig 3, the authors explain that the dispersions of the switching current are due to the density of defects, such as oxygen vacancies. Considering the structural analysis and discussions, the microstructures of both LT:HfO₂ and La:HfO₂ films might be more defective than pure HfO₂. So, their switching current must be more dispersive than pure HfO₂. Besides, there seems to be a large imprint in LT:HfO₂ films, which can lead to the difference in the switching dynamics in the positive and negative voltage regions. In Fig. 4 and the accompanying text, however, the bias polarity is not described. Another bias region of switching dynamics and the following explanation should be included.

2. The authors pointed out the enhanced ion mobility of the interstitial oxygen as the reason for the large leakage current of Ta:HfO₂. This explanation is confusing because the estimated leakage current must be ascribed to electronic carrier transport, not ion transport. Besides, the oxygen vacancies in La:HfO₂ can also increase the leakage. A more detailed explanation for the leakage should be provided.

3. In Fig. S4(f), the cycling fields for the endurance test and the information on the HZO film are missing. Also, the actual Pr (not normalized Pr) data is necessary to prove the superiority of LT:HfO₂ ultrathin films.

4. In Figure 5, switching barriers for La:HfO₂ and LT:HfO₂ were calculated; however, the doping concentrations used in the DFT calculations (12.5%) significantly exceeded the experimental values (2%). This raises concerns about the reliability of the DFT results to support the experimental data robustly. Therefore, it is recommended to conduct the calculations on 4x2x2 supercells (192 atoms), ~ 3%, to address this discrepancy. It seems that La or LT doping calculations were performed for different pathways. The authors are suggested to provide supplementary explanations for their pathway selection and elaborate on the methodology for calculating switching barriers in alternative paths.

5. The NEB results presented in Fig. 5 primarily depict homogeneous switching barriers and may not fully capture the actual switching process, which involves a small portion of domain nucleation. In light of this, it is recommended that a comparison of barriers be conducted when only a limited number of cells (e.g., 1x1x2) nucleate within a larger supercell (e.g., 4x2x2). Such an approach can potentially yield switching barriers that better align with experimental observations. Furthermore, these calculations could provide valuable insights into assessing the uniformity of switching barriers induced by La and LT doping.

6. There are several notation errors, including Supporting Information. Some are not critical, while others are critical and affect readers' understanding of authors' work.

Overall Response to Editor and Reviewers

We would like to extend our gratitude to the Reviewers for their time and efforts in evaluating our work, as well as for their pointed and valuable comments. We are encouraged by the fact that all three Reviewers found our work interesting. Also, the constructive feedbacks provided by the Reviewers have greatly contributed to improve the quality of our work. In the following sections, we address and summarize our changes made in response to the suggestions/comments from the Reviewers and provide point-by-point responses to all queries. Additionally, changes to our manuscript are highlighted in red within the revised manuscript.

Review #1

General Comment: The authors report that La/Ta-HfO₂ films can behave enhanced polarization switching behaviors due to acceptor-donor co-doping. X-ray diffraction, XPS, TEM and ferroelectric properties are investigated to verify this effective co-doping strategy. Lower concentration of oxygen vacancies can promote the synergistic improvement of ferroelectricity and higher switching speed. Although this work is interesting, the data in the present state do not fully support the claims and the novelty is insufficient, while the investigation needs to be deeper to meet the requirements of the journal

Response: We extend our appreciation to the Reviewer for his/her valuable and constructive suggestions, as well as for highlighting that "*this work is interesting*", which is totally consistent with the following two Reviewers. However, the novelty and importance of our work may not be fully appreciated yet. Below, we will address all comments in detail supported by additional experimental and theoretical evidence. Here, we first highlight the novelty and importance of this work from three aspects that make it suitable for publication in *Nature Communications*:

1. There are extremely few (if any) papers focused on understanding the root of the slow switching speed of HfO₂-based ferroelectric devices

Ferroelectric HfO₂ stands out as a highly coveted material, renowned for its excellent CMOS compatibility and scalability. However, the switchable polarization and switching speed of HfO₂ based thin films are not on par with those of perovskite ferroelectrics. The polarization switching speed of HfO₂ based ferroelectrics has emerged as a bottleneck, limiting their application in ultra-fast memory devices. Despite several papers have investigated the switching dynamics of polar HfO₂ [For instance, *Phys. Status Solidi RRL*. 15: 2000481 (2021) and *Appl. Phys. Lett.* 114 (14): 142902 (2019)], there is a notable gap in literature addressing factors influencing the switching speed of ferroelectric HfO₂ devices. This poses a genuine challenge for HfO₂-based ferroelectrics. Our research is timely and has innovatively focused on the switching behavior, examining it from the perspective of defect condition and defect-related microstructure---key factors that few researchers have explored. We propose and verify, for the first time, a model suggesting that HfO₂-based films with a less defective and more ordered microstructure favor faster switching speeds.

2. The donor-acceptor co-doping method has been successfully implemented in the realm of polar HfO₂ for the first time

In response to the defect-related slow-switching issue, the donor-acceptor co-doping concept has been creatively employed to control polar HfO₂. Utilizing cutting-edge atomic-scale imaging in transmission electron microscopy and state-of-the-art X-ray diffraction

characterizations, a comprehensive structural analysis was conducted. The results revealed that we achieved a significantly more organized and less flawed microstructure within the co-doped HfO₂ ultrathin films. Simultaneously, the co-doping method effectively modulates the concentration of oxygen vacancies. This work provides a novel approach to regulate the defect state of HfO₂-based ferroelectric materials.

3. Substantial improvements of switching performances via the co-doping method

Through systematic analysis and comparison, a notable enhancement in ferroelectricity is achieved, with a remarkable 83% increase in polarization value observed in the co-doped HfO₂ films. Concurrently, a reduction in coercive field is also achieved, representing a noteworthy discovery in HfO₂. Additionally, owing to the uniform structural state obtained in the co-doped films, robust macro-electrical characteristics, including cycling stability, persist even in films as thin as 3 nm. Most notably, a substantial enhancement in switching dynamics has been achieved, resulting in the fastest switching time ever reported among HfO₂-based ferroelectric devices. This noteworthy progress is substantiated by density functional theory calculations, which uncovered the existence of a stable intermediate configuration during polarization switching, in conjunction with a more uniform and less defective microstructure for the co-doped films. We anticipate that this work will guide and inspire the future work to pay greater attention on the microstructure and switching speed of HfO₂.

All told, our findings encapsulate a remarkable advancement in the realm of polarization switching traits within HfO₂ ferroelectric ultrathin films, encompassing aspects from polarization values to switching dynamics. Utilizing a multifaceted approach that includes meticulous advanced thin-film synthesis, cutting-edge atomic-scale imaging in transmission electron microscopy, state-of-the-art X-ray diffraction characterizations, systematic analyses of electrical properties, and density functional theory calculations, we have identified factors influence the switching speed of HfO₂-based thin films from the perspectives of microstructure, lattice configuration and switching path. Building upon these insights, we introduce a novel donor-acceptor co-doping approach utilizing La³⁺ and Ta⁵⁺ ions as acceptor and donor elements, respectively, to enhance ferroelectric properties in HfO₂ ultrathin films. In this respect, this work holds novelty, high impact, and originality.

Comment 1: Acceptor-cation doping can bring out more oxygen vacancies in HfO₂-based films. But no description about donor-cation doping was given, and more examples about this method should be provided in the introduction part.

Response: Thanks for the constructive suggestion. Following the Reviewer's suggestion, we have incorporated the description of donor-cation doping into the introduction part. Now it reads as: "Interestingly, there are relatively few studies on the ferroelectric properties of donor doped HfO₂ thin films. This is mainly because donor doping (e.g., Nb⁵⁺ doping) would suppress oxygen vacancy formation in HfO₂ and thus favor stabilizing the non-ferroelectric monoclinic phase [*J. Appl. Phys.* 122, 124104 (2017)]." (Page 3 in revised manuscript)

Comment 2: The authors only introduced the growth method of HfO₂, but no description about La:HfO₂, Ta:HfO₂ and La/Ta:HfO₂ was given. The detailed growth parameters for doping HfO₂ should also be provided. At the same time, how do the authors control the doping concentration, controlling target element or adjusting sputtering parameters?

Response: We appreciate the recommendations and suggestions to improve the manuscript. Additionally, we apologize for any confusion that may have arisen regarding to the detailed growth conditions of the HfO₂-based films. In our original manuscript, we stated that “the HfO₂-based films were grown at a temperature of 600 °C in a dynamic oxygen pressure of 15 Pa at a laser repetition rate of 2 Hz and a laser fluence of 1.35 J/cm².” It is important to note that the HfO₂-based films mentioned here encompass the La:HfO₂, Ta:HfO₂ and La/Ta:HfO₂ samples as well. That is, they are grown under identical growth condition. As suggested by the Reviewer and also to avoid further confusion, we have provided more detailed growth parameters, it now reads as: “the HfO₂-based films (including the pure HfO₂ and doped HfO₂ samples, i.e., La:HfO₂, Ta:HfO₂ and La/Ta:HfO₂ samples) were grown at a temperature of 600 °C in a dynamic oxygen pressure of 15 Pa at a laser repetition rate of 2 Hz and a laser fluence of 1.35 J/cm².” (Methods part of our revised manuscript)

The doping concentration of the films is determined by the composition of PLD ceramic targets. As suggested by the Reviewer, a complementary description is added to the Method section as below: “In addition, the composition of HfO₂-based films with different doping concentration is controlled by stoichiometric PLD targets. For instance, the 2% La:HfO₂ films are prepared using the Hf_{0.98}La_{0.02}O_{1.99} ceramic target, and the 2% LT:HfO₂ films are synthesized with the Hf_{0.96}La_{0.02}Ta_{0.02}O₂ (i.e., the La and Ta are co-doped in a 1:1 mole ratio) target. All the ceramic targets were synthesized at 1500 °C via the solid-state reaction using HfO₂ (99.99% purity), La₂O₃ (99.99% purity) and Ta₂O₅ (99.99% purity) powders.” (Methods part of our revised manuscript)

Comment 3: The authors did not give the detailed percentage of La and Ta during co-doping in the manuscript. For example, what is the atomic proportion of La and Ta in the 2% La/Ta co-doping films? Why do the authors choose this proportion? Please give more convincing evidence such as energy profile.

Response: The suggestion is appreciated as well. The detailed percentage of the co-doped films has been elucidated in response to Comment 2. Corresponding descriptions have added in the revised manuscript accordingly.

Regarding the question about the choice of a 2% La/Ta co-doping proportion, we can provide the explanation from the following perspectives. First, the selection of a 1:1 mole ratio of La and Ta is aimed at ensuring charge balance. Besides, our additional calculation results (Fig. R4) indicate that the 1:1 mole ratio favors the fastest switching process of the HfO₂-based films. Secondly, we specifically focus on the 2% LT:HfO₂ films because they exhibit the best ferroelectric properties. As shown in Fig. 3 and Fig. S6, the macro-ferroelectric behaviors of samples with different doping concentrations are exhibited. Among them, co-doped samples with a 2% doping level stand out for their highest remanent polarization.

Comment 4: Fourier transform pictures cannot be clearly seen in Figs. 1f & 1g, especially the mark of crystal orientations. Please provide pictures with better resolution.

Response: We thank the Reviewer for his/her careful reading. Figures with improved resolution have been included in our revised manuscript and Supplementary Information .

Comment 5: Dislocations are clearly seen in La:HfO₂, which may be caused by oxygen vacancies from the authors' point. However, dislocations can be generated by various factors, such as external strain, temperature and pressure etc. As can be acknowledged that tensile stress can also exist in

HfO₂-based films. Here, the authors should give more explanations to verify that dislocations are generated just because of oxygen vacancies.

Response: We are grateful to the Reviewer for bringing up this insightful comment. The Reviewer is right that dislocations can be generated by various factors, such as external strain, temperature, and pressure, etc. However, as we mentioned earlier, all the HfO₂-based films are prepared under *identical* growth conditions, including same growth temperature, oxygen pressure, cooling rate, underlying bottom electrode and substrate. In addition, the low doping level would barely affect the lattice parameters of HfO₂, and thus lattice mismatch between film and underlying substrate would remain the same, which is further supported by X-ray theta-2theta scan shown in Fig. 1a. Consequently, factors such as external strain, growth temperature, and pressure can be ruled out in our work. Therefore, the higher occurrence of dislocations in La:HfO₂ could be associated with oxygen vacancies. Previous studies [e.g., *Nat Commun.* 13, 6925 (2022); *Scr. Mater.* 188: 228-232 (2020); *J Am Ceram Soc.* 105: 1318–1329 (2022)] have demonstrated that oxygen vacancies can facilitate the nucleation of dislocations. Consequently, we attribute the dislocation generation in La:HfO₂ films to oxygen vacancies.

As suggested by the Reviewer, we have included the discussion in the revised manuscript: “Considering the identical growth condition of these doped HfO₂ films, the increased occurrence of dislocations in La:HfO₂ films is likely associated with a higher concentration of oxygen vacancies, which usually promotes the nucleation of dislocations [*Nat Commun.* 13, 6925 (2022); *Scr. Mater.* 188: 228-232 (2020)]”. (Page 6 in revised manuscript)

Comment 6: I suggest that the authors can try to quantitatively characterize the content of oxygen vacancies by STEM, EPR or XAFS.

Response: We would like to express our appreciation to the Reviewer for the suggestion. We concur with the Reviewer's assessment that determine the content of oxygen vacancies is important. However, we would also like to point out that quantitatively determining the light element content (like O) in thin films, particularly those with a thickness of only < 10 nm as in our current work, presents significant challenges (if not impossible) and is beyond our current capability. In our original manuscript, we have acquired the qualitative results of oxygen vacancies through the XPS measurement. All told, despite that quantification of oxygen vacancy content warrants further investigation, it won't affect the storyline of our current work.

Comment 7: About XPS results, the authors should provide O 1s spectra of all films and give corresponding analysis.

Response: As suggested by the Reviewer, we have included the XPS of O 1s spectra of pure HfO₂ and doped HfO₂ films in Fig. S5 of our revised Supplementary Information, as shown in Fig. R1. The core level of O 1s tested from the pure HfO₂ film is located at ~529.9 eV, with two additional peaks at ~531.4 eV and ~532.1 eV, representing absorbed oxygen and -OH [*Surf. Interface Anal.*, 26: 549-564 (1998)] respectively. Same as Hf 4f spectra, La³⁺ doped and Ta⁵⁺ doped samples exhibit contrary trends for the change of O 1s core levels. Compared with pure HfO₂, the O 1s binding energy of the Ta⁵⁺ doped sample shifts to a higher region while that of the La³⁺ doped specimen moves to a lower position. Similar phenomena have been observed in other oxides [e.g., *J. Phys. Chem. B* 109, 2445–2454 (2005) and *Adv. Mater. Technol.* 8, 2300146 (2023)]. Hence, due to the introduction of additional oxygen vacancies, La³⁺ doping results in a decrease in the energy value

of O 1s. Likewise, it is not difficult to interpret the reverse effect when Ta⁵⁺ acts as a dopant in HfO₂, where more O²⁻ are introduced and the effect of oxygen vacancies is relieved. Correspondingly, the La³⁺ and Ta⁵⁺ co-doping approach neutralizes the influence of single component doping and O 1s is virtually back the same position as that of the pure HfO₂. Corresponding analysis and spectra have been added to the revised manuscript as below:

“The O 1s spectra of La:HfO₂ and Ta:HfO₂ films present the same shift trend.” (Page 6 in the revised manuscript)

Fig. R1. the O 1s spectra of the doped HfO₂ films

Comment 8: The authors proposed that larger leakage current for Ta:HfO₂ is due to the enhanced ion mobility of interstitial oxygen. However, oxygen vacancies can also promote the increase of leakage current (See from DOI: 10.7567/JJAP.57.066501 and DOI: 10.1002/adfm.202104913). So, more explanations should be given for the main contribution of leakage current in this manuscript. Or how to distinguish these two contributions?

Response: We fully agree with the Reviewer that the leakage behavior observed in HfO₂-based films is complex. The Reviewer is correct in noting that oxygen vacancies might also promote the increase of leakage current [*Nat. Mater.* 22.5: 562-569 (2023)]. Simultaneously, previous studies [for instance, *Phys. Rev. Lett.* 89(22): 225901 (2002)] have reported that interstitial O²⁻ can also facilitate ion migration within the bulk of HfO₂ materials, leading to a significant leakage response. In essence, both types of defects, i.e., oxygen vacancies and interstitial oxygen atoms, can contribute to the increase of leakage current [*Phys. Rev. B.* 65, 174117(2002); *Phys. Rev. B.* 64, 224108 (2001)]. Based on our experimental results, the leakage current of Ta:HfO₂ thin films is nearly two orders of magnitude higher than that of the LT:HfO₂ and La:HfO₂. In our original manuscript, we attributed the larger leakage current observed in the Ta:HfO₂ film to the enhanced ion mobility of interstitial oxygen. While considering the complexity of leakage contribution and its less relevant relationship

with our storyline, we decide to omit the discussion in our original manuscript.

Comment 9: Figs. S4a & 4c and Figs. S4b & 4d, the variation tendency of polarization is not consistent with that of permittivity. Please explain or analyze the intrinsic reason.

Response: Once again, thank the Reviewer for the careful reading and constructive comment. The Reviewer is right that the variation tendency of polarization as a function of doping level does not precisely follow that of permittivity. Indeed, this phenomenon is not difficult to discern and is rather common in HfO₂-based ferroelectric films. Firstly, it is important to emphasize that higher polarization values do not guarantee higher permittivity. It is well known that non-ferroelectric tetragonal/cubic phases of HfO₂ exhibit even higher permittivity than ferroelectric orthorhombic phase [Adv. Electron. Mater. 2: 1600173 (2016)]. Secondly, similar inconsistent change of polarizations and permittivity has been frequently observed in previous studies [ACS Appl. Electron. Mater. 3, 4809, (2021), Adv. Electron. Mater. 2: 1600173 (2016)]. For example, Fig. R2a [from Adv. Electron. Mater. 2: 1600173 (2016)] vividly demonstrates the inconsistent change of polarizations and permittivity, elucidating the effect of phase transformation during cycling on polarization switching and ϵ . This is further supported by ACS Appl. Electron. Mater. 3, 4809, (2021), where a basically same tendency of polarization and permittivity is observed (Figs. R2b-c). As shown, similar to our findings, a 2% La-doped HfO₂ film exhibits largest polarization among all doped films, while the low-field permittivity of La-doped HfO₂ films simply increases with increasing doping levels.

Fig. R2. (a) The evolution of Pr and permittivity during the endurance test. The changing tendency of (b) permittivity and (c) P-E loops observed in La:HfO₂/LSMO/STO/Si (001) films with different doping concentrations

Comment 10: From Fig. 4a, it can be clearly seen that NLS model can be more appropriate to describe the switching kinetics for LT:HfO₂ film. However, KAI model is not appropriate at lower electric field. The authors should give more explanations.

Response: We sincerely appreciate the Reviewer for raising this question and apologize for any lack of clarity regarding the switching kinetics for the LT:HfO₂ film. As introduced in our original manuscript, the NLS and KAI models are two prevalent models used to describe the switching behaviors of ferroelectrics with inhomogeneous and uniform structures, respectively. The key difference between these two models lies in the distribution functions of nucleation possibility. Fig. R3 illustrates the relationship and difference of the two models. Regarding the NLS model, due to the inhomogeneity of microstructure (for instance, ferroelectric ceramics and polycrystalline films), the nucleation of reversed domains does not occur in a concerted manner, and thus the distribution function is a Lorentzian function (Fig. R3a). In contrast, the KAI model, based on the assumption

that the switching is dominated by the domain growth, has been successfully used to describe domain switching kinetics in homogeneous system, such as single crystal and epitaxial films. For the KAI model, benefiting from the uniform microstructure, almost all the domains nucleate immediately once the threshold condition is reached, ω approaches zero and the distribution function becomes a Dirac-delta function (Fig. R3b). Additionally, with an increasing applied field, the Lorentzian distribution functions become sharper, indicating that defects, impurities, and other local potential variations in inhomogeneous systems are smoothed out under sufficiently large electric fields, and we can observe a transition from NLS to KAI (Fig. R3c). Thus, the KAI model can be treated as a special case of NLS model. The Reviewer is correct that KAI model is not appropriate at low electric field (for instance, $E \leq 5$ MV/cm) for either LT:HfO₂ or La:HfO₂ films. However, when applying electric field above 5 MV/cm, the KAI model can well explain our experimental data very well for our LT:HfO₂ co-doped films. In contrast, the KAI model cannot explain the experimental data for the La:HfO₂ films even at an external field as large as 7 MV/cm, exactly due to the defect and microstructure factors, further confirming the advantage of the co-doping method.

Fig. R3. Nucleation possibility versus time for the (a) NLS model and (b) KAI model. (c) Relationship between the NLS and KAI models.

Comment 11: For DFT theoretical simulation of La/Ta-HfO₂, have the authors considered other conditions such as different proportion of La/Ta? Does there appear the lowest energy in that composition?

Response: We deeply appreciate and respect the insights provided by the Reviewer, and duly consider the suggestion. In our original manuscript, since our experiments only considered a La/Ta co-doping ratio of 1:1, so our DFT theoretical simulation of La/Ta-HfO₂ employed the same proportion. As suggested by the Reviewer, we have conducted additional DFT theoretical simulations for La/Ta-HfO₂ systems with two more La/Ta ratios, i.e., 1:2 and 2:1. A $2 \times 2 \times 2$ supercell is used to compare the polarization switching energy barrier of co-doping systems of different components. As shown in Fig. R4, the system with a La/Ta ratio of 1:1 has the lowest switching energy barrier. In short, co-doping promotes the polarization switching dynamics.

Fig. R4. DFT minimum energy paths for polarization switching in ferroelectric hafnia with different proportions of La/Ta in a $2 \times 2 \times 2$ supercell. The supercell with a La/Ta ratio of 1:1 has the lowest switching energy barrier.

Comment 12: In the “Discussion” part, the description of "co-doping can inhibit the migration of oxygen vacancies" is inaccurate, while the description of "co-doping can inhibit the generation of oxygen vacancies" seems to be more accurate in the light of the author's previous viewpoints.

Response: We express our gratitude to the Reviewer for the careful reading and insightful comment. We have done the correction accordingly in our revised manuscript.

Comment 13: In the “ferroelectric properties” section, it should be added why the ferroelectric properties of LT:HfO₂ films are substantially improved compared with that of La:HfO₂ film in the presence of a reduced concentration of oxygen vacancies.

Response: We appreciate the constructive suggestion from the Reviewer. As we demonstrate in the manuscript and above comments, we have proved that La:HfO₂ films are more defective compared to the LT:HfO₂ sample. Under such circumstances, the effect of defects, like oxygen vacancies, dislocations and other imperfections of microstructures, in the La:HfO₂ sample have a certain impact on the ferroelectric performances. We have added the discussion in our revised manuscript, it reads as: “However, the polarization value of La:HfO₂ is lower, and its switching current peaks appear blunter compared to those of LT:HfO₂. This can be ascribed to a higher defect density, including oxygen vacancies, dislocations, and microstructure distortions, which hinder the domain wall motion and consequently result in the “blunt” switching current peaks in La:HfO₂ films. **In other words, the speed of polarization reversal cannot keep up with the change in the external electric field, leading to relatively lower polarizations.** In contrast, the co-doped film exhibits a significantly enhanced polarization with sharper switching current peaks, **which benefits from the uniform microstructure with fewer defects.**” (Page 7 in the revised manuscript)

Comment 14: In the manuscript, the authors says that “this is the thinnest film whose reliable macroscopic P-E loops can be directly obtained”. However, macroscopic P-E loops were also reported in the literature (See from DOI: 10.1126/science.abm8642). Please clarify this statement.

Response: We thank the Reviewer for his/her suggestion. There are notable differences between the work (DOI: 10.1126/science.abm8642) and ours. Firstly, as depicted in Figs. R5, we conducted the

test on the 3 nm-thick sample via the conventional metal-ferroelectric-metal capacitor (MFM vertical structure). In contrast, the reference work (DOI: [10.1126/science.abm8642](https://doi.org/10.1126/science.abm8642)) obtained IDE (interdigitated electrode) result for the ultra-thin ZrO₂ film through the in-plane electrode mode, where the gap between electrodes might be micron scale (although the article presented a resistive switching loop resembling hysteresis via the vertical MFIS (metal-ferroelectric-insulator-semiconductor) structure, a macroscopic P-E loop is not provided). Secondly, due to the leakage issue and electron tunneling phenomenon, obtaining macroscopic P-E loops for the ultra-thin HfO₂ devices is challenging, and assessing the cycling endurance of such ultra-thin devices is even more difficult. Benefiting from the less defective microstructure obtained by the co-doping method in the wide bandgap material, which reduces the possibility of the tunneling behavior, we not only detect the macroscopic P-E and PUND loops of the 3 nm-thick-HfO₂ devices, but also present the cycling reliability in the Fig. S4 in our original manuscript. To our knowledge, such stable ferroelectric performances have not been reported so far.

We have cited the relevant reference indicated by the Reviewer and have corrected the sentence “this is the thinnest film whose reliable macroscopic P-E loops can be directly obtained” into “**this is one of the thinnest films whose reliable macroscopic ferroelectricity can be directly obtained [Science 376, 648-652 (2022); IEEE Electron Device Lett. 42, 1303–1306 (2021)]**” in our revised manuscript.

[Redacted]

Fig. R5. The (a) testing method and (b) corresponding result for the ultra-thin ZrO₂ film in the reference paper (DOI: [10.1126/science.abm8642](https://doi.org/10.1126/science.abm8642)). (c) The metal-ferroelectric-metal capacitor (MFM vertical structure) used for electrical characterizations in our manuscript. (d) The PUND curve of our 3 nm LT:HfO₂ film and inset is the P-E loop.

Review #2

General Comment: This is very interesting work about acceptor-donor co-doping in hafnia ferroelectric materials. Compared with La or Ta single element doping, the LT co-doped sample exhibited much better ferroelectricity and higher switching speed. These features are very important for practical memory applications. I think the overall experimental results is quite clear, however, the mechanism part still has room to improve.

Response: We thank the Reviewer for the very favorable evaluation of our manuscript and also for giving some valuable suggestions for further improving the manuscript. In the following, we carefully address all comments from the Reviewer.

Comment 1: The authors claimed the LT doped sample have much fewer defects and dislocations in structure, based on the TEM data. I suggest to check more area and larger scale in TEM. Including statistical data will make this point more solid.

Response: We appreciate the valuable suggestion from the reviewer, and we have expanded our examination to include larger-scale images. To enhance clarity, we randomly assessed the dislocation condition in four different domains and the dislocation density in each domain was calculated using the formula: $\rho = \frac{(n \times l)}{(l \times A)} = n/A$, where the n, l, A represent the number of dislocations, length of dislocation line and the area of a domain respectively. The relevant quantifiable results are presented in Fig. R6, providing an intuitive representation of higher dislocation numbers in the La:HfO₂ films. In addition, Fig.R6 has been placed in the Supplementary Information of our revised manuscript.

Fig. R6. Domain dislocation density of La:HfO₂ and LT:HfO₂ films in different regions.

Comment 2: In various literatures, La doping is an effective method to improve the HZO ferroelectricity with higher Pr, lower Ec and better endurance. However, the La doped sample did not show the same case. The process of La doping in this work may not be in the right region.

Response: We thank the Reviewer for his/her constructive suggestion. Indeed, several studies have reported that La doping can enhance the ferroelectric properties of Hf_{0.5}Zr_{0.5}O₂ films [e.g., *ACS Appl. Electron. Mater.* 2, 3221 (2020); *ACS Appl. Mater. Interfaces* 10, 2701 (2018); *Appl. Mater. Today*,

29: 101661 (2022).]. It is noted that the optimum doping level of La-doped HfO₂ would be different from La doped Hf_{0.5}Zr_{0.5}O₂. Additionally, it is crucial to acknowledge that ferroelectric HfO₂ is a metastable phase, and various factors such as strain, dopants, processing atmosphere, film thickness, electrode material, and even cooling rate, can influence the ferroelectric performances of HfO₂. Therefore, this nature determines that polar HfO₂ is highly process-dependent, and works from different groups show significant variations. Despite the complexity associated with HfO₂-based ferroelectric thin films, our experimental results are consistent with previous studies. For instance, ferroelectric La-doped HfO₂ films in previous studies [*ACS Appl. Electron. Mater.* 3, 4809 (2021); *Appl. Phys. Lett.* 120, 162904 (2022)] exhibit a similar changing tendency in ferroelectric behaviors with ours. In addition, our work and those prior studies identify the same optimum doping concentration, for La:HfO₂ films to reach the highest remanent polarization value, which ranges from 2%~3%. This consistent evolution tendency and optimum doping concentration strongly supports the feasibility and validity of our work.

Comment 3: The claim of ‘3nm film is the thinnest film whose reliable macroscopic P-E loops can be directly obtained.’ May not be appropriate. To my knowledge, ferroelectric of 1.5 nm thick HZO was directly measured by PUND in ref (IEEE Electron Device Letters 42 (9), 1303-1306).

Response: Thanks for the Reviewer’s suggestion. There are noticeable differences between our work and the study published in *IEEE Electron Device Letters* 42 (9), 1303-1306. Firstly, as illustrated in Fig. R7c, we conducted the test on the 3 nm thick sample using the conventional metal-ferroelectric-metal capacitor (MFM vertical structure). In contrast, the reference work (*IEEE Electron Device Letters* 42 (9), 1303-1306) employed an unconventional method by connecting two capacitors in series. As depicted in Fig. R7a, we have represented their principle with a simple circuit according to their design shown in the inset of Fig. R7a. The so-called in-series testing method is essentially an in-plane planar electrode device. The effective thickness of the setup is defined by the gap between the device as highlighted in Fig. R7a, which is considerably thicker than the actual film thickness. Secondly, due to the leakage issue and electron tunneling phenomenon, obtaining macroscopic P-E loops for ultra-thin HfO₂ devices is challenging, let alone assessing the cycling endurance of such devices. Benefiting from the less defective microstructure obtained by the co-doping method in the wide bandgap material, which reduces the possibility of the tunneling behavior, we not only obtained the macroscopic P-E and PUND loops of the 3 nm-thick-HfO₂ devices, but also their cycling reliability, as presented in the Fig. S6. In addition, the saturated polarization of our film is much higher than that in *IEEE Electron Device Letters* 42, 1303. To the best of our knowledge, such stable ferroelectric performances of ultrathin films (thickness less than 5 nm) have not been reported so far.

We have cited the relevant reference indicated by the Reviewer and have modified the sentence “this is the thinnest film whose reliable macroscopic P-E loops can be directly obtained” into “**this is one of the thinnest films whose reliable macroscopic ferroelectricity can be directly obtained [Science 376, 648-652 (2022); IEEE Electron Device Lett. 42, 1303–1306 (2021)]**” in our revised manuscript.

Fig. R7. (a) The test principle of the in-series testing method and (b) corresponding result. (c) The metal-ferroelectric-metal capacitor (MFM vertical structure) used for electrical characterizations in our manuscript. (d) The PUND curve of our 3 nm LT:HfO₂ film and inset is the P-E loop.

Comment 4: Pls show more evidence and analysis on the switching pathway of LT doped sample. It is far more than enough in current version.

Response: We are thankful for the insightful advice. Ferroelectric HfO₂ exhibits multiple polarization switching pathways, which have been studied in several previous works. As shown in Fig. R8, we illustrate the shift-inside (SI) mechanism during which oxygen ions move between two Hf atomic planes. Specifically, the SI-1 pathway only involves the displacement of polar oxygen ions against electric field, and the transition state is tetragonal $P4_2/nmc$ phase; the SI-2 pathway has both polar (O^p) and non-polar (O^{np}) oxygen atoms moving against the electric field, resulting in converted $O^p \rightarrow O^{np}$ and $O^{np} \rightarrow O^p$. The intrinsic switching barriers calculated with the variable-cell nudged elastic band (VCNEB) technique based on density functional theory (DFT) are 0.39 and 0.22 eV per unit cell (u.c.) for SI-1 and SI-2, respectively. In this work we focused on the SI-2 switching pathway for its lower energy barrier. We have added a paragraph discussing the switching mechanisms in Supplementary Information.

Fig. R8. Polarization switching pathways in ferroelectric HfO₂. **a** X_2^- mode in the unit cell of Pca_2_1 HfO₂ with outward- and inward-displaced oxygen atoms denoted by purple and salmon spheres, respectively. The polarization is along the z-axis. **b** Alternately arranged nonpolar oxygen ions (O^{np}) and polar oxygen ions (O^p) in Pca_2_1 HfO₂. The grey shaded area marks the polar

region. **c** Schematics of shift-inside (SI) switching pathways driven by an external electric field (E). **d** Calculated minimum energy paths for different switching pathways with DFT.

Comment 5: What is the series resistance of FeCAP, including LSMO and testing setup, the switching speed should be closely related with the series resistance. Pls also estimate the domain wall growth speed of the LT sample.

Response: We are grateful for the advice from Reviewer #2's. As is commonly known, the RC constant of the series influences the duration of the voltage loading on the ferroelectric capacitor, potentially impacting the switching dynamic measurement. To address this comment, we plan to explain it from the following two perspectives.

Firstly, we measured the RC constant of La:HfO₂ and LT:HfO₂ capacitors with different electrode areas and relevant results are shown in Fig. R9. Designed pulse was used to eliminate the polarization contribution (pre-pole and switching processes) and record the current change situation of the ferroelectric capacitor (non-switching process). As depicted in Fig. R9b, the formula $I = I_0 \times e^{-\frac{t}{RC}} + b$ (I_0 -maximum current, t -time; b -constant) is used to fit the curve and obtain the RC constants of the circuits. The RC constants of devices with diverse areas are plotted in Fig. R9c and they exhibit a linear relationship, consistent with the finding reported in previous work [*Phys. Rev. Lett.* 125(6): 067601 (2020)]. Simultaneously, the RC time varies from 89 ns to 257 ns for electrodes with a diameter ranging from 25 μm to 100 μm (the RC time of La doped and LT doped samples does not show significant differences and therefore is not discussed respectively). The RC time of these devices is considerably shorter than the switching time and does not impact the results. In addition, we have observed that the RC time of the 122.626 μm^2 ($D=12.5 \mu\text{m}$) device remains within the same order of magnitude as the switching time and we will discuss its impact on the switching time in the following section.

Fig. R9. (a) Pulse sequence used for the RC measurement. (b) The representative current signal obtain from the non-switching pulse and inset show the fitting formula and fitting result. (c) The RC time of La:HfO₂ and LT:HfO₂ capacitors with different areas.

Furthermore, we analyze the influence of the RC time on the testing results with different testing methods. Two typical pulse sequences of different methods are shown in Fig. R10. The left sequence represents a technique employing the rectangle pulse (write pulse, an ultra-fast pulse signal) and recording the switching current to explore the switching speed of a ferroelectric device. Therefore, a shorter RC constant is essential to avoid the circumstance where the charging signal of a capacitor intertwines with the polarization switching signal, which could lead to confusion in the results. However, the method adopting the right pulse sequence does not record the current signal of the ultra-fast write pulse. The read pulses detect the switchable dipoles which the write pulse reverses, avoiding the mixing of charging signal with switching signal. Despite this, the RC time still has a negative effect on the second method. When the RC time and switching time are close, the effective electric field may not be loaded in time, potentially leading to an underestimated outcome. Thus, for the 12.5 μm -diameter ferroelectric devices, the actual switching time might be faster. Nevertheless, these results do not affect the main idea of this article.

Fig. R10. Two typical pulse sequences used for the switching speed measurement.

Lastly, honestly, because of the lack of understanding about the ferroelectric domain structure within the ultrathin ferroelectric HfO_2 film which closely related to the domain wall velocity (noting that obtaining reliable PFM imaging in such a thin film is very challenging), including the lateral velocity and vertical velocity along the thickness direction, we admit that the domain wall growth speed in the ferroelectric HfO_2 ultrathin film remains unknown to us at this point.

Review #3

General Comment: Zhou et al. reported the enhanced ferroelectricity in HfO₂-based ultrathin films achieved through the acceptor-donor co-doping strategy in this manuscript. The thin films are prepared using pulsed-laser deposition on La_{0.67}Sr_{0.33}MnO₃ (LSMO)-buffered SrTiO₃ (STO) substrates. By proposing a donor (Ta⁵⁺)-acceptor (La³⁺) co-doping method, the authors reported an 83% increase in remanent polarization and robust ferroelectricity compared to the La-doped HfO₂ films, which were due to the more homogeneous and less defective microstructures. In addition, the switching kinetics measurement and first-principle calculation of the switching barrier are also performed. Overall, the paper presents the in-depth structure analysis results. Nonetheless, the technical improvement in the material properties over the previous works is marginal, significantly undermining the novelty of the work. Therefore, this work is suitable for more specialized journals.

Response: We extend our appreciation to the Reviewer for his/her valuable and constructive suggestions, as well as for highlighting that "*in-depth structure analysis*". However, the novelty and importance of our work may not be fully appreciated yet. Below, we will address all comments in detail supported by additional experimental and theoretical evidence. Here, we first highlight the novelty and importance of this work from three aspects that make it suitable for publication in *Nature Communications*:

1. There are extremely few (if any) papers focused on understanding the root of the slow switching speed of HfO₂-based ferroelectric devices

Ferroelectric HfO₂ stands out as a highly coveted material, renowned for its excellent CMOS compatibility and scalability. However, the switchable polarization and switching speed of HfO₂ based thin films are not on par with those of perovskite ferroelectrics. The polarization switching speed of HfO₂ based ferroelectrics has emerged as a bottleneck, limiting their application in ultra-fast memory devices. Despite several papers have investigated the switching dynamics of polar HfO₂ [For instance, *Phys. Status Solidi RRL*. 15: 2000481 (2021) and *Appl. Phys. Lett.* 114 (14): 142902 (2019)], there is a notable gap in literature addressing factors influencing the switching speed of ferroelectric HfO₂ devices. This poses a genuine challenge for HfO₂-based ferroelectrics. Our research is timely and has innovatively focused on the switching behavior, examining it from the perspective of defect condition and defect-related microstructure---key factors that few researchers have explored. We propose and verify, for the first time, a model suggesting that HfO₂-based films with a less defective and more ordered microstructure favor faster switching speeds.

2. The donor-acceptor co-doping method has been successfully implemented in the realm of polar HfO₂ for the first time

In response to the defect-related slow-switching issue, the donor-acceptor co-doping concept has been creatively employed to control polar HfO₂. Utilizing cutting-edge atomic-scale imaging in transmission electron microscopy and state-of-the-art X-ray diffraction characterizations, a comprehensive structural analysis was conducted. The results revealed that we achieved a significantly more organized and less flawed microstructure within the co-doped HfO₂ ultrathin films. Simultaneously, the co-doping method effectively modulates the concentration of oxygen vacancies. This work provides a novel approach to regulate the defect state of HfO₂-based ferroelectric materials.

3. Substantial improvements of switching performances via the co-doping method

Through systematic analysis and comparison, a notable enhancement in ferroelectricity is

achieved, with a remarkable 83% increase in polarization value observed in the co-doped HfO₂ films. Concurrently, a reduction in coercive field is also achieved, representing a noteworthy discovery in HfO₂. Additionally, owing to the uniform structural state obtained in the co-doped films, robust macro-electrical characteristics, including cycling stability, persist even in films as thin as 3 nm. Most notably, a substantial enhancement in switching dynamics has been achieved, resulting in the fastest switching time ever reported among HfO₂-based ferroelectric devices. This noteworthy progress is substantiated by density functional theory calculations, which uncovered the existence of a stable intermediate configuration during polarization switching, in conjunction with a more uniform and less defective microstructure for the co-doped films. We anticipate that this work will guide and inspire the future work to pay greater attention on the microstructure and switching speed of HfO₂.

All told, our findings encapsulate a remarkable advancement in the realm of polarization switching traits within HfO₂ ferroelectric ultrathin films, encompassing aspects from polarization values to switching dynamics. Utilizing a multifaceted approach that includes meticulous advanced thin-film synthesis, cutting-edge atomic-scale imaging in transmission electron microscopy, state-of-the-art X-ray diffraction characterizations, systematic analyses of electrical properties, and density functional theory calculations, we have identified factors influence the switching speed of HfO₂-based thin films from the perspectives of microstructure, lattice configuration and switching path. Building upon these insights, we introduce a novel donor-acceptor co-doping approach utilizing La³⁺ and Ta⁵⁺ ions as acceptor and donor elements, respectively, to enhance ferroelectric properties in HfO₂ ultrathin films. In this respect, this work holds novelty, high impact, and originality.

Comment 1: In Fig 3, the authors explain that the dispersions of the switching current are due to the density of defects, such as oxygen vacancies. Considering the structural analysis and discussions, the microstructures of both LT:HfO₂ and La:HfO₂ films might be more defective than pure HfO₂. So, their switching current must be more dispersive than pure HfO₂. Besides, there seems to be a large imprint in LT:HfO₂ films, which can lead to the difference in the switching dynamics in the positive and negative voltage regions. In Fig. 4 and the accompanying text, however, the bias polarity is not described. Another bias region of switching dynamics and the following explanation should be included.

Response: We appreciate the constructive comment to improve the manuscript further. The dispersive switching current of pure HfO₂ is due to the metastable nature of the polar phase. As discussed in our manuscript, the orthorhombic *Pca2*₁ ferroelectric phase of HfO₂ is metastable and generally does not form under conventional growth conditions, unless a large number of defects, such as oxygen vacancies, are introduced. Consequently, compared to LT:HfO₂ and La:HfO₂, the pure HfO₂ film is predominantly composed of the non-ferroelectric monoclinic phase, resulting in a weak ferroelectric response, as shown in Fig. 1a and Fig. 3a in our manuscript. Additionally, the coexistence of multiple phases appears to be more prevalent in pure HfO₂ films. These factors likely contribute to the dispersive switching current in pure HfO₂ film.

The Reviewer is correct that imprint would lead to differences in the switching dynamics in the positive and negative voltage regions. As suggested by the Reviewer, we have carried out additional experiments to investigate the switching dynamics in the negative voltage region for LT:HfO₂ and La:HfO₂ films (Fig. R11). As shown in Fig. R11, the switching time of the LT:HfO₂

device is almost 1/10 of that of the La:HfO₂ sample, further highlighting the advantage of the co-doping approach. Moreover, the negative bias switching kinetics demonstrate the same behavior as observed in the positive voltage region.

Following the Reviewer's suggestion, we have included the Fig. R11 in Supplementary Information of our revised manuscript, and added the text in our Supplementary Information, it reads as "Due to the asymmetric electrode (LSMO/doped-HfO₂/Pt) configuration, a notable imprint can be observed in La:HfO₂ and LT:HfO₂ systems. The influences of imprint on the switching dynamics are also investigated. A pulse sequence with opposite directions to the designed pulses shown in Fig. S7 is employed to explore the switching behavior in the negative voltage region. As displayed in Fig. S9, the changing tendency of the switching models in the negative region shows no difference compared to that in the positive region, and the differences in switching speed between them are even more prominent during the negative switching process. The negative voltage switching process further demonstrates the superiority of the La-Ta co-doping method."

Fig. R11. The reverse switching dynamics test of the (a) LT:HfO₂ and (b) La:HfO₂ films.

Comment 2: The authors pointed out the enhanced ion mobility of the interstitial oxygen as the reason for the large leakage current of Ta:HfO₂. This explanation is confusing because the estimated leakage current must be ascribed to electronic carrier transport, not ion transport. Besides, the oxygen vacancies in La:HfO₂ can also increase the leakage. A more detailed explanation for the leakage should be provided.

Response: Thanks for this comment. We totally agreed with the Reviewer that the leakage behavior observed in HfO₂-based films is complicated. The Reviewer is correct that oxygen vacancies could also increase the leakage [Nat. Mater. 22.5: 562-569 (2023)]. At the same time, previous studies [for instance, Phys. Rev. Lett. 89(22): 225901 (2002)] also reported that interstitial O²⁻ could also facilitate the ion migration in the bulk of HfO₂ materials, leading to large leakage response. In other words, both two types of defects, i.e., oxygen vacancies and interstitial oxygen atoms (which can act as traps for electrons and holes, respectively), could contribute to an increase of leakage current [Phys. Rev. B. 65, 174117(2002); Phys. Rev. B. 64, 224108 (2001)]. From our experimental results, the leakage current of Ta:HfO₂ thin films is nearly two orders of magnitude higher than that of the LT:HfO₂ and La:HfO₂. Therefore, in our original manuscript, we attributed the larger leakage current observed in the Ta:HfO₂ film to the enhanced ion mobility of interstitial oxygen. Since the leakage mechanism of the Ta:HfO₂ film is not directly pertinent to the main theme of our current work, and considering the complexity of leakage contribution, we have decided to omit the

discussion about the leakage response of the Ta:HfO₂ film in our original manuscript.

Comment 3: In Fig. S4(f), the cycling fields for the endurance test and the information on the HZO film are missing. Also, the actual Pr (not normalized Pr) data is necessary to prove the superiority of LT:HfO₂ ultrathin films.

Response: We are again extremely thankful to the Reviewer for his/her very constructive comment. Following the Reviewer's suggestion, we have provided details about the cycling fields for the endurance test and the information on the HZO film in our revised Supplementary Information, it now reads as "All samples are cycled with the 100 kHz bipolar rectangular wave and read with the 5 kHz triangle wave. Among them, the 6-nm-thick films are cycled and read under a 4 V voltage and the 3-nm-thick capacitors are cycled and read with a 2.5 V voltage" (Page 4 in the Supplementary information)

In addition, the graph of actual Pr versus the cycle numbers is provided below, and the corresponding figure has been updated in the Supplementary Information. We appreciate the suggestion once again.

Fig. R12. The endurance test result of these doped HfO₂-based film.

Comment 4: In Figure 5, switching barriers for La:HfO₂ and LT:HfO₂ were calculated; however, the doping concentrations used in the DFT calculations (12.5%) significantly exceeded the experimental values (2%). This raises concerns about the reliability of the DFT results to support the experimental data robustly. Therefore, it is recommended to conduct the calculations on 4x2x2 supercells (192 atoms), ~3%, to address this discrepancy. It seems that La or LT doping calculations were performed for different pathways. The authors are suggested to provide supplementary explanations for their pathway selection and elaborate on the methodology for calculating switching barriers in alternative paths.

Response: We thank Reviewer for this insightful suggestion. Ferroelectric HfO₂ has multiple polarization switching pathways, which have been studied in several previous works. As shown in Fig. R8, we illustrate the shift-inside (SI) mechanism during which oxygen ions move between two Hf atomic planes. Specifically, the SI-1 pathway only involves the displacement of polar oxygen ions against electric field, and the transition state is tetragonal *P4₂/nmc* phase; the SI-2 pathway has both polar (*O^p*) and non-polar (*O^{np}*) oxygen atoms moving against the electric field, resulting in converted *O^p* → *O^{np}* and *O^{np}* → *O^p*. The intrinsic switching barriers calculated with the variable-cell nudged elastic band (VCNEB) technique based on density functional theory (DFT) are 0.39 and

0.22 eV per unit cell (u.c.) for SI-1 and SI-2, respectively. In this work we focused on the SI-2 switching pathway for its lower energy barrier. We have added a paragraph discussing the switching mechanisms in Supplementary Information. We note that La and LT doping calculations were performed for the exact same type of pathway, SI-2.

Following the suggestions, we have performed additional calculations using $4 \times 2 \times 2$ supercells, with results shown in Fig. R13. It is evident that the co-doped system has a much lower switching barrier than the La doped system.

Fig. R13. Energy barriers of SI-2 switching pathway in La:HfO₂ and LT:HfO₂ using $4 \times 2 \times 2$ supercells.

Comment 5: The NEB results presented in Fig. 5 primarily depict homogeneous switching barriers and may not fully capture the actual switching process, which involves a small portion of domain nucleation. In light of this, it is recommended that a comparison of barriers be conducted when only a limited number of cells (e.g., $1 \times 1 \times 2$) nucleate within a larger supercell (e.g., $4 \times 2 \times 2$). Such an approach can potentially yield switching barriers that better align with experimental observations. Furthermore, these calculations could provide valuable insights into assessing the uniformity of switching barriers induced by La and LT doping.

Response: We are thankful for the Reviewer's suggestion. Our computational model was established based on experimental results. As detailed in the manuscript, the switching dynamics of the LT:HfO₂ system can be fitted with KAI model, which is used to describe the switching behavior characterized by concentrated nucleation but limited by domain expansion. Therefore, we mainly focused on a homogeneous switching model to simulate the polarization switching. Moreover, our calculations revealed that the large switching barrier in La:HfO₂ is mainly due to the migration of oxygen vacancies, a local effect that may also manifest when a small number of cells nucleate within a larger supercell as suggested by the Reviewer.

Comment 6: There are several notation errors, including Supporting Information. Some are not critical, while others are critical and affect readers' understanding of authors' work.

Response: We thank the Reviewer for his/her careful reading. Following the Reviewer's suggestion,

we have double checked the spelling carefully and corrected all the typos/notation errors in the manuscript, including Supplementary Information. We have incorporated these corrections in the revised manuscript.

REVIEWER COMMENTS

Reviewer #1 (Remarks to the Author):

The authors have given corresponding responses to most suggestions, but there still exist some questions to be further clarified. Although there are numerous experimental results, the internal physical mechanism is not clear enough. Below is a list of my concerns:

1. For the XPS spectra of O-1s in doped HfO₂ films, the authors attributed the small shoulder of O-1s to the existence of absorbed oxygen and -OH. Actually, XPS measurement should be done after etching on surfaces, which can avoid the influence of these factors. Meanwhile, some investigations considered this part to be the non-lattice oxygen or oxygen vacancies. so, I think the conclusion about XPS is not convincing.
2. As mentioned by two reviewers, oxygen vacancy is a major contribution to the large leakage current. The authors have considered that interstitial O²⁻ is the main factor for explanation of large leakage. The authors should give more evidence and analysis on this contribution, such as high-resolution atomic phase structure. Actually, the authors have proposed that La&Ta co-doping can suppress the formation of oxygen vacancies in the introduction part, why do you think the theme of current work is not related to oxygen vacancy? It is quite confusing.
3. Dislocations can be clearly seen from HRTEM images in Fig. 1(f) and Fig. S3(e), the authors should also consider the changes of crystalline structure, for example, the calculation of content of o-phase and t-phase.
4. For Fig. S4, I can not see the area of region A, B, C and D in the HRTEM images. Please recheck!
5. The authors have only given PUND curves of LT co-doped HfO₂ films. The corresponding PUND measurements for Pure HfO₂, La:HfO₂ and also Ta:HfO₂ should also be provided. To my knowledge, coercive field can also represent the switching capabilities of films. So, the coercive field for the above samples should also be compared and corresponding analysis should be given.

Reviewer #2 (Remarks to the Author):

The authors have addressed my comments appropriately. It can be published now.

Reviewer #3 (Remarks to the Author):

Although some queries raised in the original manuscript have been resolved, several ambiguous aspects still remain, as shown below.

1. In Fig. S6, the La-doped HfO₂ samples have an overall higher permittivity than the LT co-doped HfO₂ samples. As the authors replied, the permittivity of the films can vary depending on the effects of the non-ferroelectric phases or the dead layer effect. If so, is there structural data proving that the La-doped samples have higher permittivity phases (tetragonal or cubic) than the LT co-doped samples?
2. Although the discussion on the leakage current has been omitted in the revised manuscript, several issues remain unclear. The authors attributed the higher leakage current of the Ta-doped HfO₂ films to the increased ion mobility. If true, ions move inside the film

and cause damage such as ion scattering. Cycling characteristics generally deteriorate when thin films are damaged by the scattering effects. Including this leakage perspective, the endurance data of Ta-doped HfO₂ must be provided to ensure the superiority of the LT co-doping method.

3. Fig. S14 shows that the switching barrier of SI-1 (≈ 0.35 eV/f.u. ≈ 1.4 eV/u.c.) is higher than that of SI-2 (≈ 0.2 eV/f.u. ≈ 0.8 eV/u.c.). The authors stated that the switching pathway of SI-1 is related to P42/nmc, but they used the switching pathway of SI-2 in the calculation of switching barriers of La-HfO_{2-x} and LT-HfO₂, according to Fig. 5. However, these results are controversial with the previous study (see Fig. 1 (f) in Mater. Today 50, 8–15 (2021)) reporting that the switching pathway of P42/nmc has the lowest barrier of ≈ 0.4 eV/u.c. This previous study indicates that the switching barrier can vary significantly depending on which switching pathway is selected. In addition, the calculated values in this study show significant differences with the previous study. Therefore, all the possible switching pathways of La-HfO_{2-x} and LT-HfO₂ should be systemically calculated in comparison with pure HfO₂. Then, the lowest switching barriers among the switching pathways should be compared to demonstrate that the switching barrier is lowered by the co-doping effect.

4. The authors did not answer the comment 5 of previous reviewer #3 by claiming the homogeneous nucleation. As described above, however, the ferroelectric hafnia has lots of switching pathways. In addition, several previous studies (Materials Today Physics 34, 101064 (2023), Phys. Rev. Lett. 131, 226802 (2023)) have reported that the switching barrier varies significantly depending on whether the pathway is homogeneous (using $1 \times 1 \times 1$ u.c.) or nucleation/domain wall growth ($N \times 1 \times 1$ supercells) switching. Also, the previous work (Figs. 5 (b) and (c) in Materials Today Physics 34, 101064 (2023)) already described that both the nucleation and domain wall propagation barriers in La-doped HfO₂ are significantly lowered than in undoped-HfO₂. These results show the variance of switching barriers depending on switching pathways. Thus, exploration of the switching pathways is needed to elucidate the difference in switching kinetics of La-HfO_{2-x} and LT-HfO₂.

5. For the above comments on the calculation parts, this reviewer questions whether the oxygen vacancy present in La-HfO_{2-x} plays a role in lowering the switching speed compared to LT-HfO₂. Rather than the effect of oxygen vacancy, the difference in phase fraction between LT-HfO₂ and La-HfO_{2(-x)} can affect the electrical properties such as switching speed and uniformity. In addition to the permittivity difference mentioned in comment 1, LT-HfO₂ shows higher $2P_r$ than La-HfO₂ (Fig. 3 (c)). These results indicate that LT-HfO₂ has more o-phase than La-HfO_{2(-x)}.

Overall Response to Editor and Reviewers

We would like to express our sincere appreciation to the Reviewers for dedicating their time and efforts to evaluate our work and for providing valuable feedback. We have thoroughly reviewed all the comments and have made revisions accordingly. In the following section, we address and summarize our changes made in response to the suggestions/comments from the Reviewers and provide point-by-point responses to all queries. Additionally, changes to our manuscript are highlighted in red within the revised manuscript.

Review #1

General Comment: The authors have given corresponding responses to most suggestions, but there still exist some questions to be further clarified. Although there are numerous experimental results, the internal physical mechanism is not clear enough.

Response: We would like to extend our gratitude to the Referee once again for providing multiple insightful and constructive comments. That feedback has been helpful for improving our manuscript to the current point. Below, we address each comment individually.

Comment 1: For the XPS spectra of O-1s in doped HfO₂ films, the authors attributed the small shoulder of O-1s to the existence of absorbed oxygen and -OH. Actually, XPS measurement should be done after etching on surfaces, which can avoid the influence of these factors. Meanwhile, some investigations considered this part to be the non-lattice oxygen or oxygen vacancies. so, I think the conclusion about XPS is not convincing.

Response: Thanks for the valuable suggestions. Firstly, we fully acknowledge the complexity associated with interpreting the XPS spectra of O 1s in metal oxides (Noting that both absorbed oxygen and -OH belong to non-lattice oxygen). Indeed, several studies have attributed the intermediate-energy feature, typically located around 530 eV~532 eV, to oxygen vacancies, for instance, in *Nanoscale*. 12, 9024 (2020) (Fig. R1a) and *ACS Nano*. 18, 1214 (2024) (Fig. R1b). However, we believe that such explanation is probably inappropriate. If we consider the intermediate peaks originated from oxygen vacancies, it is abnormal that the oxygen vacancy signal is even much stronger than the lattice oxygen peak, suggesting that there are more oxygen vacancies than lattice oxygen atoms within the solid-state oxides. Therefore, just as depicted in Fig. R1c-d, some reports [*Chem. Mater.* 35, 5468 (2023); *Surface Science*. 712: 121894 (2021); *J. Am. Chem. Soc.* 146, 1, 1071–1080, (2024).] favor the opinion that the peak cannot stand for the oxygen vacancies, and it is more reasonable to attribute the “oxygen vacancy” region signal to other non-lattice oxygen, such as absorbed water molecules or surface oxygen passivated with hydrogen, which is totally consistent with ours.

Meanwhile, since the signal comes from the weakly-bonded oxygen atoms, hydroxyl, and water, these signals are more vulnerable to external factors such as the atmosphere, exposure time and others. In view of this, we carried out additional XPS measurements by comparing the spectra of the LT:HfO₂ films, which are freshly prepared (as-grown sample), exposed to the atmosphere (293 K and 30% RH air) for half a year, and after surface-etching (Ar plasma, 0.23 nm/s etching rate and 3s etching time), respectively (other doped samples are same and we take the LT:HfO₂ sample as an example). The corresponding results are shown in Fig. R2. The O 1s spectra intuitively demonstrates the effect of time and treatment conditions on the XPS results. For a more direct comparison for the changes in peak conditions, we treat the signal of absorbed oxygen atoms,

hydroxyl and water as one peak here. The signal located between 530 eV and 532 eV increases significantly as the time extends, while it decreases dramatically after etching. The phenomenon further supports that the changeable peak isn't an intrinsic reflection of the material feature and cannot represent oxygen vacancies. The non-lattice oxygen, including adsorbed oxygen, hydroxyl, or water, fit well with our experimental results. In addition, the peak position, FWHM of the lattice oxygen signal barely changed, further supporting that our XPS results are valid.

[Redacted]

Fig. R1. Some O1s analysis in literatures. (a) a schematic of oxygen vacancies in a RuO₂ film deposited on TiN [*Nanoscale*. 12, 9024-9031,(2020)]. (b) the O 1s analysis of the catalysts [*ACS Nano*. 18, 1, 1214–1225, (2024)]. (c) an explanation for the O 1s [*Chem. Mater.* 35, 14, 5468–5474 (2023)] and (d) the O1s spectra of the CeO_{2-x} catalysts [*J. Am. Chem. Soc.* 146, 1, 1071–1080, (2024)].

Fig. R2. The O1s spectra of the LT:HfO₂ film which is (a) freshly prepared, (b) exposed to the air for half a year and (c) Ar plasma etched.

Overall, the Reviewer is correct in pointing out the lack of a consensus on the interpretation of XPS O 1s spectra. As suggested, we fully acknowledge this aspect. Therefore, we have re-plotted the original fitting peaks of adsorbed oxygen and hydroxyl into the non-lattice oxygen peak (Fig. R3) and added it to the revised supplementary information (Fig. S6).

Fig. R3 The O 1s spectra of doped HfO₂ films.

Comment 2: As mentioned by two reviewers, oxygen vacancy is a major contribution to the large leakage current. The authors have considered that interstitial O²⁻ is the main factor for explanation of large leakage. The authors should give more evidence and analysis on this contribution, such as high-resolution atomic phase structure. Actually, the authors have proposed that La&Ta co-doping can suppress the formation of oxygen vacancies in the introduction part, why do you think the theme of current work is not related to oxygen vacancy? It is quite confusing.

Response: We also apologize for any confusion that may have arisen regarding the interpretation of oxygen vacancy-related phenomena. Characterizing point defects such as oxygen vacancies or interstitials presents significant challenges and is currently beyond our capabilities. Even with state-of-the-art (S)TEM imaging techniques, directly observing single point defects remains difficult. Instead, we acknowledge the various possibilities and the complexity of the leakage mechanism in HfO₂-based dielectric films. Following the Reviewer's suggestion, we have taken a closer look at the conduction mechanism in our samples. Despite our careful and detailed study, our current manuscript is not intended to be the final word. There are still open questions regarding the true physics of such systems that warrant further investigation.

In ferroelectric oxide thin films, the leakage mechanism depends not only on the defect state of the ferroelectric materials, but also on factors such as film thickness, film-electrode interface, etc. Therefore, the effect of oxygen vacancy depends on the specific conduction mechanism (Fig. R4). Defects, such as oxygen vacancies, typically act as traps of carriers. For bulk-limited mechanisms, charge carriers jump from traps to the conduction band of the dielectrics (like the Poole-Frenkele mechanism) or tunnel across the dielectrics via traps (like the Hopping conduction), contributing to the leakage current. Under such circumstances, as the Reviewers proposed, oxygen vacancies could act as the major source of leakage response. While defects do not always play a role of the "bad

guys” for the resistivity [*Adv. Mater.* 26, 6341 (2014) and *Adv. Mater.* 28, 10750 (2016)]. For example, defects like lead vacancies act as deep electron traps (with trap energies of 0.4-0.5 eV) and enhanced, rather than reduce, the electric resistivity of PbTiO₃ films [*Adv. Mater.* 28, 10750 (2016)].

In the interface-limited conduction mechanism, the interface barrier height Φ_B plays a significant role. In the Schottky emission mechanism, the change in current density J follows the relationship:

$$J = A^* T^2 \exp \left[\frac{-q(\Phi_B - \sqrt{qE/4\pi\epsilon_{opt}\epsilon_0})}{k_B T} \right] \quad (1)$$

Where A^* is the effective Richardson constant, E is the electric field across the dielectric, k_B is the Boltzmann’s constant, T is the temperature and ϵ_{opt} is the optical dielectric constant. The barrier height Φ_B at the metal-dielectric interface can be reduced by the image force, enabling carriers to move from the electrode to the dielectric (Fig. R4b), resulting in the rising of leakage current. Meanwhile, the state of dielectric (conduction type) and electrodes (metal with different work functions or P/N -type semiconductor) can cause additional effects on the energy barrier.

Fig. R4 (a) A typical classification of leakage mechanisms in dielectric films; (b-e) Schematics of energy band diagrams for the common conduction mechanisms.

As suggested by the Reviewer, we further investigated the doping effect on the leakage mechanisms of the HfO₂-based ferroelectric thin film. We studied the current density and applied electric field (J - E) relations of the LSMO/doped-HfO₂/Pt capacitors for both positive and negative bias in the temperature range from 350 K to 450 K, as depicted in Fig. R5. The leakage current densities of all films noticeably increase with rising temperatures, with the Ta-doped HfO₂ films exhibiting the highest leakage current densities across all temperatures. Meanwhile, as shown in Fig. R6, a straight line can be fitted to the $\ln(J/T^2)$ versus $E^{1/2}$ plot, and the optical dielectric constants ϵ_{opt} of HfO₂ extracted from the slopes are within the range of ~4-6, which are close to the expected value [*Ceram. Int.*, 47, 33751 (2021)], meaning that the leakage response of the doped HfO₂ films obey the Schottky emission mechanism.

Fig. R5. The leakage current conditions of (a) La:HfO₂, (b) LT:HfO₂ and (c) Ta:HfO₂ films at different temperatures.

Fig. R6. Diagrams of the (a) positive and (b) negative bias of measured currents. And the Schottky emission fitting results of (c) La:HfO₂, (d) LT:HfO₂ and (e) Ta:HfO₂ devices.

To further elucidate the nature of conduction behavior, the Schottky barrier heights Φ_B are further extracted from the intercepts of lines in Fig. R7 according to the formula (1) and the literature [Appl. Phys. Lett. 90, 072902 (2007)]. As shown, the linear dependence further validates the Schottky emission mechanism. Both barrier heights Φ_B of LSMO/doped-HfO₂ and Pt/doped-HfO₂ interfaces of the La:HfO₂ and LT:HfO₂ are higher than those of the Ta:HfO₂ sample, directly explaining the smaller leakage current observed in the La doped and LT doped devices. Corresponding analysis has been added to the revised Supplementary Information (Fig. S8).

Fig. R7 The interface energy barrier of the (a) La:HfO₂, (b) LT:HfO₂ and (c) Ta:HfO₂ devices respectively.

In our previous response, we did not state that our current work is not related to oxygen vacancies, we affirm that our results indicate a significant correlation between oxygen vacancies and microstructural inhomogeneity, which adversely affects polarization switching. Specifically, oxygen vacancies hinder polarization switching, leading to low polarization values and sluggish switching performance. Expanding on these insights, we propose a novel donor-acceptor co-doping strategy employing La^{3+} and Ta^{5+} ions as acceptor and donor elements, respectively, to improve the ferroelectric properties of HfO_2 ultrathin films. This co-doping approach resulted in significantly improved microstructure organization and reduced flaws within the HfO_2 ultrathin films. Concurrently, the co-doping method effectively modulates the concentration of oxygen vacancies. Notably, a remarkable 83% increase in polarization value is observed in the co-doped films, accompanied by a reduction in coercive field. Most notably, a substantial enhancement in switching dynamics has been obtained, manifesting in the impressively fastest switching process ever reported among HfO_2 -based ferroelectric devices.

Comment 3: Dislocations can be clearly seen from HRTEM images in Fig. 1(f) and Fig. S3(e), the authors should also consider the changes of crystalline structure, for example, the calculation of content of o-phase and t-phase.

Response: Thanks for the constructive suggestion. Following the advice from the Reviewer, we closely examined the XRD patterns presented in Fig. 1a in the manuscript and carefully analyzed the diffraction peaks of doped- HfO_2 . As depicted in Fig. R8, the main peaks situated in the position ranging from 27° to 32° can be divided into two peaks, representing the Monoclinic and Orthorhombic/Tetragonal phases, respectively. The area of different peaks can qualitatively represent the content of different structures, and the area ratios are summarized in Fig. R8b. A large amount of Monoclinic phase was detected in the pure and Ta-doped HfO_2 films, while it is rare in the La- and LT-doped films. Also, the main peaks of La- and LT-doped samples appear essentially the same. However, the main reflections of the o-phase and t-phase are very close, and the diffraction peaks are broad due to the very low film thickness. Therefore, identification of orthorhombic and tetragonal phases in the La- and LT- doped HfO_2 films is very challenging. Due to the similarity between the O phase and T phase, the phase content cannot be directly determined from the X-ray diffraction or TEM information. Such difficulties are discussed, for example, in previous work published by Schroeder et al., *JJAP* 58, SL0801 (2019). For these reasons, we, like other groups in the field, combine XRD, TEM with other techniques to discern between the existing phases. The additional techniques used in our work, such as dielectric and ferroelectric properties measurements and analysis, are also commonly employed in the field.

It is well known that the dielectric response of ferroelectrics is determined by both intrinsic and extrinsic contributions [*J. Electroceram.* 19: 49-67, (2007)], and the intrinsic permittivity can be a good indicator of phase. The intrinsic contribution originates from the lattice response of individual domains, while the extrinsic part is primarily associated with the motion of interfaces such as domain walls and phase boundaries. The extrinsic contribution becomes negligible at high DC electric fields, allowing the intrinsic contribution to be isolated. Therefore, the permittivity under large DC bias, where extrinsic response like domain wall/phase boundary motion is suppressed, is treated as the intrinsic permittivity of the ferroelectric material [like *Science*. 381,558-563, (2023)]. Therefore, to further investigate the phase fraction of La: HfO_2 and LT: HfO_2 samples, we conducted dielectric-frequency spectra measurements with a large DC bias of 3 V. As shown in

Fig. R8 (a) Analysis of phase condition of doped-HfO₂ films. (b) The area ratio of different phases extracted from XRD patterns.

Fig. R9a, nearly identical dielectric responses were observed for the La-doped and LT-doped samples (2%, 6 nm), implying similarity in the phase structure of the La- and LT-doped films. Additionally, frequency-dependent P-E loops with a fixed testing electric field are presented in Fig. R9b to support this phenomenon. A decrease in polarization values with increasing frequency is observed in both the La-doped and LT-doped capacitors. It is a common phenomenon as the speed of domain expansion cannot catch up with the change of the applied electric field when the frequency rises. Despite that, there are two points worth noting. One is that the polarization value

Fig. R9. (a) The dielectric-frequency spectra of the doped HfO₂ with a fixed DC bias of 3V. (b) The frequency-dependent P-E loop measurement with a fixed electric field for the La-HfO₂ and LT-HfO₂ ferroelectric capacitors. (c) The ϵ -V curves of 6 nm La doped devices and LT doped samples with different doping concentrations (the yellow dashed lines mark the intrinsic permittivity). (d) A brief schematic of the phase evolution trend for HfO₂ films with different thicknesses and doping concentrations.

of the samples prepared via La-doping and LT-codoping are close when the frequency are sufficient low (500 HZ, providing enough time for dipoles to switch), signifying the similar phase fraction and approximately the same dipoles within their structures, which coincides with the same permittivity situation. Another point to note is the changing tendency of the remanent polarization (P_r) values. Apparently, the performance of the La-doped sample is highly frequency-dependent, while the LT-doped sample is less affected, which demonstrates the reason why the La-HfO₂ shows lower $2P_r$ than the LT-HfO₂ and further proves negative effects brought by oxygen vacancies.

As demonstrated above, it can be inferred that the La:HfO₂ (2%, 6 nm) film and LT:HfO₂ (2%, 6 nm) sample exhibit the similar phase fraction. However, upon closer examination of the ϵ -V curves for different doping concentrations, an abrupt increase in permittivity can be observed for the La-doped device with a doping concentration of 4%, indicating a partial transformation from the polar Orthorhombic phase (PO phase) to a higher permittivity phase. Oxygen vacancies, introduced by La doping, are conducive to pushing the polar phase towards a higher symmetry paraelectric phase. In contrast, the LT 4% sample maintain a stable ϵ , which can be attributed to the co-doping technique reducing the presence of oxygen vacancies and thus suppressing the phase transformation. Meanwhile, the ultra-thin film thickness of 6 nm, which takes advantage of the “reverse-size effect” feature [*Nature*, 580(7804): 478-482, (2020)] of HfO₂ based ferroelectric films, ensures that our epitaxial La (2%) and LT (2%) doped films mainly consist of polar orthorhombic phase. Hence, as revealed above, combining the dielectric response, ferroelectric properties, and structural information, a brief evolution path can be summarized in Fig. R9d, and the La:HfO₂ (2%, 6 nm) film and LT:HfO₂ (2%, 6 nm) sample exhibit similar phase fractions and mainly consist of the polar orthorhombic phase.

Based on this reality, we acknowledge our inappropriate labeling in Fig. S3e in the Supplementary Information, and we have updated the O/T marks to O. Also, corresponding descriptions in SI have been changed.

Comment 4: For Fig. S4, I cannot see the area of region A, B, C and D in the HRTEM images. Please recheck!

Response: We thank the Reviewer for the careful reading. The regions previously designated as Region A~D have been relabeled as Region I to IV to avoid confusion with the sequence numbers used to describe domain variants. These regions (marked as 1 (I), 2 (II), 3 (III), and 4 (IV)) are utilized to denote different areas for the statistical analysis of domain dislocation densities. As shown in Fig. R10, the high-resolution STEM images have been processed using the Inverse Fast Fourier Transform (IFFT) method to visually represent the dislocation situations (the effectiveness of the method is validated by Fig. R10a, where the defective state becomes more apparent after the IFFT treatment). The number of dislocations observed in the La:HfO₂ samples is notably higher than that in the co-doped samples.

Following the Reviewer’s suggestion, we have added the analysis results into our revised Supplementary Information Fig. S4.

Fig. R10. (a)-(d) Region 1 to region 4 for the La:HfO₂ sample. (e)-(h) Region I to region IV for the LT:HfO₂ sample. The marker “⊥” is used to symbol the dislocation.

Comment 5: The authors have only given PUND curves of LT co-doped HfO₂ films. The corresponding PUND measurements for Pure HfO₂, La:HfO₂ and also Ta:HfO₂ should also be provided. To my knowledge, coercive field can also represent the switching capabilities of films. So, the coercive field for the above samples should also be compared and corresponding analysis should be given.

Response: We thank the Reviewer for the constructive comments. As suggested by the Reviewer, the PUND curves and corresponding changes in remnant polarizations and coercive fields of these HfO₂-based films are presented in the Fig. R11 (and Supplementary Information Fig. S7). From the PUND measurements, the co-doped sample exhibits the highest polarization value ($2P_r$) of 34 $\mu\text{C}/\text{cm}^2$, which is significantly higher than that of the La doped samples, consistent with the results from dynamic P - E loop measurements. In contrast, the pure HfO₂ and Ta:HfO₂ films exhibit lower P_r values due to the presence of a certain amount of paraelectric phase (seen in Fig. 1a). The coercive

fields of these samples are also analyzed. As shown in Fig. R11b, the coercive fields (E_C) of the pure HfO_2 and Ta:HfO_2 samples are similarly smaller than those of the La-doped and LT-doped samples. The La-doped sample exhibits the largest E_C among all these samples. Besides the higher content of polar phase, factors like oxygen vacancy-pinned domain wall and microstructure distortion also contribute to the increased coercive field. In contrast, although the P_r value of the co-doped film is largest among all the samples, the E_C of the film is smaller than that of the La-doped one. The phenomenon further demonstrates the superiority of the co-doping method in reducing the switching barrier and accelerating the switching process.

Fig. R11. (a) PUND curves of HfO_2 -based capacitors with different doping conditions. (b) The changing tendency of remnant polarizations and coercive fields along with different doping situations.

Review #2

General Comment: The authors have addressed my comments appropriately. It can be published now.

Response: We sincerely express our gratitude to the Reviewer for affirming our work. We believe that our study can serve as a timely report and provide guidance for efforts aimed at improving the switching behaviors of hafnia-based ferroelectrics.

Review #3

General Comment: Although some queries raised in the original manuscript have been resolved, several ambiguous aspects still remain.

Response: We thank the Reviewer for his/her favorable comments and constructive suggestions. In the following we address each of them individually.

Comment 1: In Fig. S6, the La-doped HfO₂ samples have an overall higher permittivity than the LT co-doped HfO₂ samples. As the authors replied, the permittivity of the films can vary depending on the effects of the non-ferroelectric phases or the dead layer effect. If so, is there structural data proving that the La-doped samples have higher permittivity phases (tetragonal or cubic) than the LT co-doped samples?

Response: We appreciate the Reviewer for the insightful comment. First of all, under the circumstance of the near-identical interfaces, film thicknesses, and device configuration, the permittivity could be a good indicator of the phase content. However, there might be some misunderstandings of the permittivity for the Reviewer. It is well known that the dielectric response of ferroelectrics is determined by both intrinsic and extrinsic contributions [*J. Electroceram.* 19: 49-67, (2007)]. The intrinsic contribution originates from the lattice response of individual domains, while the extrinsic part is primarily associated with the motion of interfaces such as domain walls and phase boundaries. The extrinsic contribution is negligible at high dc electric fields and the intrinsic contribution can be obtained subsequently. Therefore, the permittivity under large DC bias, where extrinsic response like domain wall/phase boundary motion is suppressed, is treated as the intrinsic ϵ of the ferroelectric material [like *Science.* 381,558-563, (2023)]. Hence, as indicated by the yellow dotted line in the Fig. R12a, the ϵ under high-bias field of these two series of doped HfO₂ do not show big differences, especially when the doping range is between 1%~3%. This is further supported by the dielectric-frequency spectra measurement with a large DC bias of 3 V (Fig. R12b). As is shown, nearly the same dielectric response of the La-doped and LT-doped samples (2%, 6 nm) is observed, implying the similarity of phase structure.

Fig. R12. (a) The ϵ -V curves of La-doped and LT-doped samples with different doping concentrations. The yellow dotted lines mark the values of permittivity of every capacitor. (b) The dielectric-frequency spectra of the doped HfO₂ with a fixed DC bias of 3V.

Meanwhile, following the Reviewer's suggestion, the structural analysis can equally support

our opinion. We take a close look at XRD patterns shown in Fig. 1a in the manuscript and carefully analyze the diffraction peaks of doped-HfO₂. As shown in Fig. R13, main peaks located in the position ranging from 27° to 32° can be divided into two peaks, which represent the Monoclinic and Orthorhombic/Tetragonal phases respectively. The area of different peaks can qualitatively represent the content of different structures and the area ratios are summarized in Fig.R13b. Considerable Monoclinic phase can be detected in the pure and Ta doped HfO₂ films, which is rare in the La and LT doped films. Also, the main peak of La and LT doped samples exhibit almost the same diffraction condition. Combing the dielectric response, ferroelectric properties, and structural information (Fig. R9), it can be judged that the La:HfO₂ and LT:HfO₂ films exhibit similar phase fractions and mainly consist of the polar Orthorhombic phase.

Fig. R13. (a) Analysis of phase condition of doped-HfO₂ films. (b) The area ratio of different phases extracted from XRD patterns.

Comment 2: Although the discussion on the leakage current has been omitted in the revised manuscript, several issues remain unclear. The authors attributed the higher leakage current of the Ta-doped HfO₂ films to the increased ion mobility. If true, ions move inside the film and cause damage such as ion scattering. Cycling characteristics generally deteriorate when thin films are damaged by the scattering effects. Including this leakage perspective, the endurance data of Ta-doped HfO₂ must be provided to ensure the superiority of the LT co-doping method.

Response: We thank the Reviewer for his/her valuable suggestion. In response to Comment 2 from Reviewer #1, we conducted additional experiments to elucidate the leakage mechanism, finding that the leakage conduction is primarily governed by interface-limited Schottky emission. Furthermore, our results indicated that the Ta:HfO₂ sample demonstrates the lowest Schottky barrier height among all doped HfO₂ samples. This elucidates why the Ta:HfO₂ sample exhibited the highest leakage current density. As suggested by the Reviewer, we have included the endurance behavior of Ta:HfO₂ films, which is depicted in Fig. R14. Consistent with the Reviewer's expectations, the Ta:HfO₂ sample exhibits poor cycling performance attributed to severe leakage issues and a large amount of non-ferroelectric phase. Due to the substantial leakage current observed in Ta:HfO₂ devices during the cycling test, the P_r of Ta:HfO₂ samples appear bigger, which is also demonstrated in the Fig. 3a in the manuscript). The updated endurance data has been added to our revised Supplementary Information (Fig. S7).

Fig. R14. The endurance behaviors of these HfO₂-based devices with different doping conditions and thicknesses

Comment 3: Fig. S14 shows that the switching barrier of SI-1 (~ 0.35 eV/f.u. = ~ 1.4 eV/u.c.) is higher than that of SI-2 (~ 0.2 eV/f.u. = ~ 0.8 eV/u.c.). The authors stated that the switching pathway of SI-1 is related to P4₂/nmc, but they used the switching pathway of SI-2 in the calculation of switching barriers of La-HfO_{2-x} and LT-HfO₂, according to Fig. 5. However, these results are controversial with the previous study (see Fig. 1 (f) in Mater. Today 50, 8–15 (2021)) reporting that the switching pathway of P4₂/nmc has the lowest barrier of ~ 0.4 eV/u.c. This previous study indicates that the switching barrier can vary significantly depending on which switching pathway is selected. In addition, the calculated values in this study show significant differences with the previous study. Therefore, all the possible switching pathways of La-HfO_{2-x} and LT-HfO₂ should be systemically calculated in comparison with pure HfO₂. Then, the lowest switching barriers among the switching pathways should be compared to demonstrate that the switching barrier is lowered by the co-doping effect.

Response: We appreciate this important comment. We apologize for a typo in the y-axis label of Figure S14(d). The correct unit should be 'eV/u.c.' instead of 'eV/f.u.', as now accurately labeled in Fig. R15(d). Therefore, the switching barrier for the SI-1 pathway from our DFT calculations is 0.34 eV/u.c., consistent with the previous study (~ 0.4 eV/u.c.). Similarly, the barrier for the SI-2 pathway is 0.22 eV/u.c., which is lower than that for the SI-1 and SA pathways.

We concur with the Reviewer that “the switching barrier can vary significantly depending on the chosen switching pathway.” Following this suggestion, we have performed additional calculations for various SI-1 and SI-2 pathways in La:HfO_{2-x} and LT:HfO₂. As demonstrated in Fig. R16, the SI-2 pathway in LT-HfO₂ exhibits the lowest barrier, further corroborating our main conclusion that acceptor-donor co-doping can improve the ferroelectric switching properties. This figure is now added as Fig.S17 in Supplementary Information.

Fig. R15. Unit-cell-level polarization switching pathways in ferroelectric HfO₂. [Adapted from PRL

131, 256801 (2023)] (a) X_2^- mode in the unit cell of $Pca2_1$ HfO_2 . (b) Alternately arranged nonpolar oxygen ions (O^{np}) and polar oxygen ions (O^p) in $Pca2_1$ HfO_2 . (c) Schematics of shift-inside (SI) and shift-across (SA) switching pathways driven by an external electric field (ϵ). (d) Calculated minimum energy paths for different switching pathways using the DFT-based variable-cell NEB method.

Fig. R16. Switching barriers for the SI-1 and SI-2 pathways in $La:HfO_{2-x}$ and $LT:HfO_2$. A $2 \times 2 \times 1$ supercell is used.

Comment 4: The authors did not answer the comment 5 of previous reviewer #3 by claiming the homogeneous nucleation. As described above, however, the ferroelectric hafnia has lots of switching pathways. In addition, several previous studies (Materials Today Physics 34, 101064 (2023), Phys. Rev. Lett. 131, 226802 (2023)) have reported that the switching barrier varies significantly depending on whether the pathway is homogeneous (using $1 \times 1 \times 1$ u.c.) or nucleation/domain wall growth ($N \times 1 \times 1$ supercells) switching. Also, the previous work (Figs. 5 (b) and (c) in Materials Today Physics 34, 101064 (2023)) already described that both the nucleation and domain wall propagation barriers in La-doped HfO_2 are significantly lowered than in undoped- HfO_2 . These results show the variance of switching barriers depending on switching pathways. Thus, exploration of the switching pathways is needed to elucidate the difference in switching kinetics of $La-HfO_{2-x}$ and $LT-HfO_2$.

Response: We thank Reviewer for this insightful suggestion. In response to the previous comment, we have compared various pathways for homogenous switching. The conclusion is that the SI-2 pathway has the lowest barrier, which can be further reduced by La-Ta co-doping.

The two articles cited by the Reviewer focus on the motion of the $Pbca$ -type 180° domain wall. Both studies reported that the “crossing” mechanism, which can be viewed as the SA pathway at the domain wall, has a lower barrier than the “non-crossing” mechanism. We have reproduced their results, as shown in Fig. R17(a): the crossing pathway is energetically favored. We then investigate the crossing mechanism in $La:HfO_{2-x}$ and $LT:HfO_2$ using $8 \times 1 \times 1$ supercells. As demonstrated in Fig. R17(c), the co-doping slightly reduces the barrier compared to that in $La:HfO_{2-x}$. This figure is now included as Fig.S18 in the supplementary information. Importantly, we note that the barrier is ~ 0.35 eV for lateral motion by one unit cell in $LT:HfO_2$, which compares unfavorably with the homogenous SI-2 pathway with a barrier of 0.14 eV/u.c. (see Fig. R16). Unlike perovskite ferroelectrics, our results suggest that homogenous switching is kinetically favored over the motion

of the *Pbca*-type 180° domain wall.

[Redacted]

Fig. R17. Energy barriers for a *Pbca*-type domain wall motion process in (a) pure HfO₂ and (c) La:HfO_{2-x} and LT:HfO₂. Because the non-crossing path has a high barrier, we only investigate the crossing path in doped HfO₂. The results from [*Materials Today Physics* 34, 101064 (2023)] are presented in (b) and (d) for comparison. An 8×1×1 supercell is used in our calculations. It is noted that we introduced oxygen vacancy in La-doped HfO₂ to balance the charge, which explains the subtle difference between (c) and (d) for the crossing paths in La-doped HfO₂.

Comment 5: For the above comments on the calculation parts, this reviewer questions whether the oxygen vacancy present in La-HfO_{2-x} plays a role in lowering the switching speed compared to LT-HfO₂. Rather than the effect of oxygen vacancy, the difference in phase fraction between LT-HfO₂ and La-HfO_{2-x} can affect the electrical properties such as switching speed and uniformity. In addition to the permittivity difference mentioned in comment 1, LT-HfO₂ shows higher 2Pr than La-HfO₂ (Fig. 3 (c)). These results indicate that LT-HfO₂ has more O-phase than La-HfO_{2-x}.

Response: We thank the Reviewer for the constructive comment. We have addressed the phase fraction circumstance in detail in response to Comment 1, and we will address Comment 5 from the following two aspects. First, as pointed out by Reviewer #3, the phase fraction can affect switching behaviors such as switching speed. Additionally, as expounded in the paper [*Adv. Electron. Mater.* 8, 2100420,(2022)], the non-polar phase indeed has a positive effect in suppressing the pinned domain propagation, which favors better endurance and faster switching performances of the HZO ferroelectric capacitors. Therefore, the La-doped sample should work better than the LT-doped device if there are more other non-polar phases in the films prepared by La doping. However, the experimental result is contrary to the expectation, indicating that the phase fraction of the La-doped

and LT-doped films may not differ significantly. Secondly, apart from the dielectric test we have explained in the comment 1, frequency-dependent ferroelectric measurements are also implemented. Frequency-dependent P-E loops with a fixed testing electric field are shown in Fig. R18. Decreased polarization values along with increasing frequency can be observed in both the La-doped and LT-doped capacitors. This phenomenon is common as the speed of domain expansion cannot keep pace with the change of the applied electric field when the frequency rises. Despite that, there are two points worth noting. One is that the polarization value of the samples prepared via the La-doping and LT-doping is similar when the frequency are sufficiently low (500 Hz), providing enough time for dipoles to switch. This similarity signifies a comparable phase fraction and approximately the same dipoles within their structures, which coincide with the similar permittivity situation. Another point worth noting is the changing tendency of P_r values. Apparently, the performance of the La-doped sample is highly frequency-dependent, while the LT-doped sample acts less affected. This difference demonstrates why La-HfO₂ shows lower $2P_r$ than the LT-HfO₂ and further confirms negative effects brought by oxygen vacancies. To intuitively compare the difficulty of switching, another frequency-dependent P-E loop measurement with fixed P_r values is carried out simultaneously. As displayed in Fig. R19a, a larger coercive field can be detected with increasing frequency for both the La:HfO₂ and LT:HfO₂ devices, illustrating that it becomes harder to switch dipoles at higher frequencies. The frequency-dependent coercive fields (Fig. R19b) follow the rule that $E_C \propto f^\beta$, which is consistent with the report [Sci Rep. 4772 (2014)]. Simultaneously, compared with the steep change of E_C extracted from the La:HfO₂ ferroelectric capacitor at different frequency, the E_C of the LT:HfO₂ sample exhibits a gradual change with rising frequency. This phenomenon highlights the effect of oxygen vacancies in pinning the domain wall and switchable dipoles.

Fig. R18. The frequency-dependent P - E loop measurement with a fixed drive electric field for the La-HfO₂ and LT-HfO₂ ferroelectric capacitors.

Fig. R19. (a) The frequency-dependent P - E loop measurement with a fixed Pr value for the La-HfO₂ and LT-HfO₂ ferroelectric capacitors. (b) the change tendency of coercive fields for the La-doped and LT-doped samples with different frequencies.

REVIEWER COMMENTS

Reviewer #1 (Remarks to the Author):

The authors have provided excellent responses to the suggestions, however there exist some minor problems to be further clarified before publication in this highly renowned journal. Below is a list of my concerns:

1. About the explanation of Response 1, it is my opinion that the shoulder of O-1s can be attributed to non-lattice oxygen including absorbed water molecules, surface oxygen passivated with hydrogen and oxygen vacancy. How do you distinguish their contributions to the performance of HfO₂-based thin films?
2. There is a wrong annotation in Figure R8a. Please recheck.
3. For Figure R9s, the authors have only given permittivity-voltage curves which however are not dielectric-frequency spectra. So, there is no relation between Figure R9b and 9c.
4. A small concern, what is the design ideas for the experiments of La&Ta co-doping? Will co-doping of other two elements with different valences generate similar experimental effects?

Reviewer #3 (Remarks to the Author):

The authors clarified all issues raised in previous rounds. Therefore, this paper deserves publication in Nature Communications.

Overall Response to Editor and Reviewers

We sincerely express our gratitude to the Editor and Reviewers for their valuable time and meticulous efforts in reviewing our manuscript. We have thoroughly reviewed all the comments and have made revisions accordingly. In the following section, we address and summarize the changes made in response to the suggestions/comments from the Reviewers and provide point-by-point responses to all queries.

Review #1

General Comment: The authors have provided excellent responses to the suggestions, however there exist some minor problems to be further clarified before publication in this highly renowned journal.

Response: We would like to extend our appreciation to the Referee for providing multiple insightful and constructive comments. The feedback has significantly contributed to the improvement of our manuscript to its current state. In the subsequent sections, we comprehensively address any remaining concerns.

Comment 1: About the explanation of Response 1, it is my opinion that the shoulder of O-1s can be attributed to non-lattice oxygen including absorbed water molecules, surface oxygen passivated with hydrogen and oxygen vacancy. How do you distinguish their contributions to the performance of HfO₂-based thin films?

Response: We appreciate the reviewer for once again bringing this to our attention. We fully agree with the Reviewer's opinion regarding the XPS shoulder of O-1s. Meanwhile, surface electrochemistry of ferroelectric films does have effects on the electric properties, especially for ultrathin films. For example, one very recent study by Sergei V. Kalinin *et al.* has demonstrated that surface electrochemical state would play an important role on phase stability and polarization switching behavior of Hf_{0.5}Zr_{0.5}O₂ thin films [*Nat. Mater.* 22, 1144–1151 (2023)]. Therefore, investigating effects of surface condition on HfO₂-based thin films is important and warrants further investigation. However, in our work, all the films are prepared and kept with the same conditions and related devices are fabricated with the newly as-prepared films. The surface conditions are basically the same for our samples. Moreover, distinguishing the contributions of various surface condition on ferroelectric performances presents significant challenges and needs lots of extra experiments, which is currently beyond our capabilities.

Comment 2: There is a wrong annotation in Figure R8a. Please recheck.

Response: We apologize for the typo we have made in the last Response letter and thanks for the careful reading. The annotation has been corrected and is shown in the Fig. R1 below.

Fig. R1 Analysis of phase condition of doped-HfO₂ films.

Comment 3: For Figure R9s, the authors have only given permittivity-voltage curves which however are not dielectric-frequency spectra. So, there is no relation between Figure R9b and 9c.

Response: We appreciate the Reviewer's suggestions. The dielectric-frequency spectra is given in the Fig. R9a in the last Response letter. The relationship between Fig.R9a (the dielectric-frequency spectra) and R9c (permittivity-voltage curve) in the last Response letter can be depicted in the Fig. R2.

As shown in Fig. R2s, both electric signals used for the ϵ -V and ϵ -f spectra can be divided into two parts: DC electric bias/field and AC excitation bias/field (used to detected the capacitance). For the ϵ -V spectra measurement, the signal is a changing DC bias coupled with a small AC bias with a given fixed frequency (Fig. R2a), whereas signal used for the ϵ -f spectra measurement is AC signals with changing frequencies superimposed on a fixed background DC bias (Fig. R2b). Therefore, the relationship between ϵ -V and ϵ -f spectra is vividly. We can directly infer the intrinsic dielectric response under high DC bias from ϵ -V curves and roughly extract the phase information. Similarly, we could also use the ϵ -f measurement to measure the intrinsic dielectric response by measuring the dielectric response under high DC biases, where the large DC bias acts to suppress extrinsic response, such as domain wall motion.

Fig. R2. (a) The signal used for permittivity-voltage measurement. Inset is a typical ϵ -V curve. (b) The signal used for dielectric-frequency experiments. Inset is a ϵ -f curve.

As regards Fig. R9b (P-E loops measured with pulses of different frequency) in the last Response letter, the frequency-dependent P-E loops further explained the pinning effect brought by oxygen vacancies in the La:HfO₂ device. We clarify our opinion from multiple perspectives such as

dielectric and ferroelectric aspects, and they are complementary to each other.

Comment 4: A small concern, what is the design ideas for the experiments of La&Ta co-doping? Will co-doping of other two elements with different valences generate similar experimental effects?

Response: Thanks for the constructive comment. As we described in the Introduction part in our manuscript, oxygen vacancies are helpful to stabilize the metastable polar phase of HfO₂, yet their side effects on switching characteristics are gradually recognized. The double-edged sword effect brought by oxygen vacancies is an urgent matter must be overcome in exploring high-performance hafnium-based devices. Some attempts have been tried like two steps method [*IEEE Electron Device Letters*, 43(7): 1057-1060, (2022)]. Our donor-acceptor co-doping method offer a direct method to regulate defect conditions and it turns out to be a useful way to improve the switching characteristics (like switching speed, polarizations and coercive fields) of HfO₂-based devices. We believe that the donor-acceptor co-doping method can be generally applicable to other system and similar experimental effects can be expected in other co-doping system, like Nb⁵⁺-La³⁺ and Ta⁵⁺-Y³⁺, co-doped HfO₂ thin films.

Review #3

General Comment: The authors clarified all issues raised in previous rounds. Therefore, this paper deserves publication in Nature Communications.

Response: We would like to express our gratitude to the Reviewer once again for providing us with valuable comments to enhance the manuscript and for recommending it for publication.

REVIEWERS' COMMENTS

Reviewer #1 (Remarks to the Author):

The manuscript has been well modified according to the suggestions, which can fullfill the publication in this journal.

Overall Response to Editor and Reviewers

We sincerely express our gratitude to the Editor and Reviewers again for their valuable time and meticulous efforts in reviewing our manuscript. In the following section, we address and summarize the changes made in response to the suggestions/comments from the Reviewers and provide point-by-point responses to all queries.

Review #1

General Comment: The manuscript has been well modified according to the suggestions, which can fulfill the publication in this journal.

Response: We would like to extend our appreciation to the Reviewer once again for providing us with valuable comments to enhance the manuscript and for recommending it for publication.